# Ozone Saline Solution Polarizes Microglial Cells Towards an Anti-Inflammatory Phenotype

**DOI:** 10.3390/molecules30193932

**Published:** 2025-09-30

**Authors:** Federica Armeli, Beatrice Mengoni, Martina Menin, Gregorio Martínez-Sánchez, Mauro Martinelli, Maurizio Maggiorotti, Rita Businaro

**Affiliations:** 1Department of Medico-Surgical Sciences and Biotechnologies, Sapienza University of Rome, 04100 Latina, Italy; federica.armeli@uniroma1.it (F.A.); beatrice.mengoni@uniroma1.it (B.M.); menin_martina@yahoo.it (M.M.); 2Independent Researcher, 60126 Ancona, Italy; gregorcuba@yahoo.it; 3Ozone Therapy Unit, Department of Biomedical Sciences, Ospedale San Pietro Fatebenefratelli, 00189 Rome, Italy; mauro.martinelli53@gmail.com; 4Italian College of Oxygen-Ozone Therapy (CIO3), 00186 Rome, Italy; maggiorotti@gmail.com

**Keywords:** ozone therapy, ozonated saline solution, microglia, anti-inflammatory response, *Nrf2*, reactive oxygen species, cytokines, immunomodulation

## Abstract

Ozone (O_3_) therapy has demonstrated antioxidant and anti-inflammatory properties, but the systemic administration of ozonated saline solution (O_3_SS) remains underexplored. This study evaluates the cytotoxicity, antioxidant response, and immunomodulatory effects of O_3_SS on murine microglial (BV2) and human endothelial (HUVEC) cells. Cells were exposed to increasing doses of O_3_ (1, 5, or 10 μg/NmL) dissolved in saline. Viability assays showed that low doses (1 and 5 μg/NmL) enhanced cell proliferation without cytotoxicity, while the highest dose (10 μg/NmL) reduced viability and increased cell death. O_3_SS treatment upregulated antioxidant genes, including *Nrf2* and *SOD1*, and decreased reactive oxygen species in lipopolysaccharide (LPS)-stimulated microglia. Additionally, O_3_SS modulated microglial phenotype by reducing pro-inflammatory markers (*iNOS*, *IL-1β*) and increasing anti-inflammatory markers (*Arg-1*, *IL-10*). Immunofluorescence confirmed enhanced Arg-1 protein expression, indicating a shift toward an anti-inflammatory state. These results suggest that low-dose O_3_SS activates cellular antioxidant defenses and promotes an anti-inflammatory microglial phenotype, supporting its potential as a safe systemic O_3_ therapy. Further studies are warranted to confirm in vivo efficacy and optimize clinical protocols.

## 1. Introduction

Ozone (O_3_) molecules, formed by three oxygen atoms, are highly unstable. They can quickly break down into one O_2_ molecule and one single oxygen atom, which acts as a strong oxidant. O_3_ is formed in the atmosphere as a result of electrical discharges and UV light during thunderstorms [1]. In the stratosphere (between 15 and 35 km above sea level), its function is protective, as it shields the Earth from harmful UV rays. In contrast, at the level of the troposphere (10–15 km from the ground), O_3_ is produced due to the action of nitrogen oxides and volatile organic pollutants. Together with these pollutants, O_3_ causes damage to the respiratory system, predisposing individuals to the development of even serious pathologies [1]. Due to its strong oxidizing properties, it was widely used during the World Wars for wound treatment and disinfection, and it continues to be

Employed today for water sanitation [2]. The biological effects of O_3_ are concentration-dependent and follow a hormetic pattern. While high concentrations (>80 µg/mL) are toxic and harmful, low concentrations within the well-defined therapeutic window (1–80 µg/mL) are beneficial. This beneficial effect is mediated through the stimulation of cellular antioxidant defenses, which are ultimately capable of counteracting oxidative stress, as detailed below [3]. In this context, O_3_ has attracted significant interest due to its anti-inflammatory and antioxidant properties, placing it within the scope of complementary medicine—able to support the treatment of certain pathologies without fitting the definition of a pharmaceutical [4]. These characteristics form the basis of O_3_ therapy [5,6,7]. Preclinical studies using in vitro cellular models have elucidated the pathways underlying the protective actions of O_3_, and numerous reports have highlighted the therapeutic efficacy of O_3_ in pain relief, in the treatment of herniated disks, diabetic foot, osteoarthritis, and various other pathologies of the musculoskeletal system [8]. O_3_ is administered in a variety of ways, from topical application for the treatment of skin lesions and oral diseases, to systemic application, for anti-inflammatory and anti-aging, and pro-longevity treatment. For the latter, a mixture of O_3_ and oxygen is added to a certain amount of blood taken from the patient and then reinfused back to the patient. This method, although widely used, has several limitations, as it is not always well accepted by patients: it requires large-caliber venous access, and is sometimes refused due to some religious beliefs [9]. It has therefore been proposed to use ozonized saline solution (O_3_SS) in which the gas is bubbled and which is then administered to patients via intravenous drip [10].

The aim of this study was to evaluate the cytotoxicity, the antioxidant response, and the immunomodulatory effects of O_3_SS on murine microglial cells (BV2) and human endothelial cells (HUVEC). Specifically, we sought to assess the toxic effects of different concentrations of O_3_ dissolved in saline and explore the antioxidant and anti-inflammatory responses triggered by O_3_ exposure. These effects are of particular interest given the potential use of ozone therapy for treating central nervous system disorders. To address these objectives, we employed two complementary in vitro models. Primary cultures of human endothelial cells were used to evaluate potential cytotoxic effects that may manifest in vivo, specifically as phlebitis. Additionally, a mouse microglial BV2 cell line was utilized to investigate the potential anti-inflammatory as well as antioxidant activity of O_3_, given the growing interest in O_3_ therapy for neurodegenerative diseases. A key aspect of this study was to determine the minimum effective O_3_ concentration that would elicit beneficial antioxidant and anti-inflammatory responses without inducing cytotoxicity. Previous studies have shown that ozone autohemotherapy can provide therapeutic benefits in diseases like acute cerebral infarction [11], with evidence supporting its ability to increase Nrf2 nuclear translocation and the expression of key antioxidants such as SLC7A11 and GPX4 in models of cerebral ischemia/reperfusion injury [12]. Furthermore, ozone therapy has demonstrated its potential in improving the outcomes of central nervous system diseases, particularly in the treatment of ischemic stroke and Alzheimer’s disease. O_3_ can effectively reduce inflammation, modulate oxidative stress signals, lower amyloid β peptide levels, and improve tissue oxygenation [13]. A further study concluded that Oxygen-O_3_ therapy would contribute to improving Multiple Sclerosis (MS) patients by elevating the Treg cell responses [14]. In addition, it is known that the brain microenvironment is shaped by glial cells receiving information from neurons, vascular elements, as well as by peripheral immune signals, providing the rationale for the use of O_3_ for the treatment of central nervous system disorders [15]. In this context, our goal was to explore whether O_3_ exposure may modulate microglial activation states and stimulate endogenous antioxidant defenses. Through these models, we aimed to determine whether the O_3_SS mixture acts as a modulator of cellular antioxidant pathways and inflammatory responses, thereby supporting its potential use in diseases characterized by oxidative stress and inflammation [16].

## 2. Results

In the present preclinical study, we aimed to evaluate the efficacy of low doses of O_3_SS added to cultures of BV2 microglial cells in carrying out an antioxidant and anti-inflammatory activity, after having evaluated their possible cytotoxic activity.

As shown in Figure 1A, 24 h after O_3_ treatment at a concentration of 1 or 5 μg/NmL in saline solution, a significant proliferative response in BV2 cell cultures was detected. On the contrary, the O_3_ at a concentration of 10 μg/NmL significantly reduced the number of live cells and increased the number of dead cells. The same doses applied to primary human endothelial cell cultures (HUVECs) are non-cytotoxic and significantly increase the number of viable cells. In contrast, O_3_ at concentrations of 5 and 10 μg/NmL leads to an increase in the number of dead cells (Figure 1B).

To rule out that the cytotoxic effects observed in Figure 1 were due to the formation of hypochlorite (ClO^−^) during the ozonation process, we measured its concentration in the ozonated solutions using a specific colorimetric kit. The measured hypochlorite concentration was negligible (0.001%) across all O_3_ concentrations tested, a value similar to that found in the culture medium alone and corresponding to the detection limit of the assay kit. This result indicates that hypochlorite formation was not a significant factor contributing to the cytotoxicity of the O_3_ solutions.

The activation of cellular antioxidant pathways in response to treatment with O_3_ dissolved in 0.9% saline solution was evaluated by the analysis of several mRNAs, respectively, after 4 and 24 h from treatment, as shown in Figure 2. To investigate the molecular mechanism underlying the antioxidant effects of O_3_SS, we assessed the activation of the Nrf2 pathway by measuring the expression of Nuclear factor erythroid 2-related factor 2 (*Nrf2*) and Superoxide Dismutase 1 (*SOD1*) mRNA in BV2 cells after 4 or 24 h: after 4 h only the dose of 5 μg/NmL was effective in inducing an increase in *Nrf2* mRNA, while doses of 1 and 10 caused a decrease. The results changed at 24 h where a significant increase in *Nrf2* mRNA expression was observed with both doses of 5 and 10 μg/NmL. *SOD1* mRNA increased both at 4 and 24 h following BV2 treatment with 5 or 10 μg/NmL O_3_.

We then evaluated the expression of Reactive Oxygen Species (ROS) in BV2 microglial cells after treatment with LPS, which is known to induce significant oxidative stress in innate immunity cells, in the presence or absence of O_3_. As illustrated in Figure 3, the amount of ROS was decreased in the presence of O_3_ treatment in 0.9% saline solution.

We also evaluated the ability of O_3_ treatment to reverse the pro-inflammatory phenotype (namely M1 phenotype) induced by the addition of LPS to BV2 microglia cells. We analyzed the expression of mRNAs of *iNOS* (inducible Nitric Oxide Synthase) and *Arg-1* (Arginase-1), two specific markers of the M1 pro-inflammatory phenotype and the M2 anti-inflammatory phenotype, respectively (Figure 4). O_3_ at 5 μg/NmL was shown to be able to decrease the expression of *iNOS* mRNA, as measured 4 h after cell treatment; 24 h after treatment, the effective dose was found to be 10 μg/NmL. O_3_ administered with saline solution was able to increase the expression of *Arg-1*, a marker of anti-inflammatory polarized microglial cells, at all the tested concentrations, after 4 h from the treatment. In the presence of LPS best results were obtained with 1 and 5 μg/NmL O_3_ dissolved in saline solution.

The anti-inflammatory activity of O_3_ treatment was confirmed by immunofluorescence experiments (Figure 5) where O_3_SS added at the concentrations of 1 and 5 μg/NmL induced the expression of the protein ARG-1 also in the presence of LPS, counteracting the cell polarization towards a pro-inflammatory phenotype.

Furthermore, to confirm the anti-inflammatory activity of O_3_SS added to microglial cells, we evaluated the expression of mRNA for the pro-inflammatory cytokine *IL-1β* both in the absence and presence of LPS (Figure 6A). As shown in Figure 6A, O_3_ treatment, at all tested concentrations, induced a decrease in *IL-1β* mRNA expression both 4 h and 24 h after treatment. In the presence of LPS, O_3_ pretreatment was effective in decreasing *IL-1β* mRNA expression at O_3_ concentrations of 5 and 10 μg/NmL; after 24 h from the LPS treatment, all O_3_ concentrations were effective in inducing a decrease in *IL-1β* mRNA. We also evaluated the ability of O_3_SS to switch microglia cells towards a reparative, anti-inflammatory phenotype (M2) by assessing the expression of the anti-inflammatory cytokine *IL-10* (Figure 6B). O_3_ at concentrations of 1, 5, 10 μg/NmL was able to upmodulate the expression of *IL-10* mRNA 4 h after treatment, as shown by RT-PCR experiments. At the same time, it was seen whether this activity was also present after the addition of LPS. After 4 h no significant variations were appreciated, which instead emerged 24 h after treatment; as a matter of fact, an upmodulation of *IL-10* mRNA was detected both at 1 and 5 μg/NmL O_3_, confirming the anti-inflammatory activity of O_3_ treatment.

## 3. Discussion

Our study investigated the response of two distinct cellular models—BV2 microglial cells and HUVEC cells—following exposure to O_3_-saturated 0.9% saline solution (O_3_SS), prepared by bubbling O_3_ for 20 min to achieve final concentrations of 1, 5, or 10 µg/NmL. Our results demonstrate that the O_3_SS does not present toxic activity in itself in the tested concentration ranges. Previous studies have analyzed the possible reactions taking place after the addition of O_3_ to saline solutions, ten ding with cytotoxicity. In this connection, Razumovski et al. (2010) [17] established that O_3_ decomposition processes in NaCl aqueous solutions are not accompanied by the formation of noticeable amounts of hypochlorites and chlorates. In addition, Peritiagyn demonstrated that the concentration of sodium hypochlorite in the O_3_SS was less than 0.001 g/mL [18]. We have confirmed this result by the hypochlorite detection kit. Moreover, it was demonstrated that ozonation of the saline solution eliminates traces of Bromine that exist in the normal pharmacological formulation of the saline [19]. The kinetics of O_3_ saturation in the saline solution were previously studied using spectrophotometric methods. At the concentrations used and with the same medical device and conditions, the O_3_ concentration in the solution stabilizes after 10 min of bubbling and corresponds to 10% of the initial concentration. Under these conditions, the formation of hypochlorous acid or hydrogen peroxide was not detected [20]. Traces of Fe^2+^ in sodium chloride can, in principle, catalyze hydroxyl radical (•OH) generation through the Fenton reaction. However, this requires the presence of H_2_O_2_ or another suitable peroxide. In pure pharmaceutical-grade NaCl solution (0.9% saline), the European pharmacopeia sets a maximum Fe content of 2 ppm, and under standard ozonation conditions (1–10 µg/mL O_3_), no detectable H_2_O_2_ is formed. Analytical studies confirm H_2_O_2_ levels remain far below the threshold necessary for Fenton chemistry. Therefore, while Fe^2+^ traces could act as catalysts in theory, in practice under these conditions the formation of hydroxyl radicals via the Fenton reaction is ruled out [21]. The proportion of O_3_SS added to the DMEM medium was 1:20. Since DMEM contains a buffering system, significant pH variations after the addition of O_3_SS were not expected.

Recently, systemic O_3_ application using O_3_SS was found to be well tolerated, with no serious adverse events reported in the preliminary study. The intervention, performed under controlled conditions and at low doses (3 μg/NmL), did not result in clinically significant toxicity or hematological abnormalities. Importantly, the study observed a transient increase in the survivor of circulating CD34+ cells following systemic O_3_ administration, suggesting a potential stimulatory effect on endothelial–hematopoietic stem/progenitor cell mobilization [22], perhaps explaining the proliferative effect observed on HUVEC cells in our results.

These findings are consistent with the established concept that low-dose O_3_ can induce a mild, controlled oxidative eustress, which in turn activates endogenous antioxidant and regenerative pathways without causing cellular damage [23,24]. The observed mobilization of CD34+ cells aligns with the literature describing O_3_’s capacity to enhance tissue regeneration and modulate immune responses through redox bioregulation [7].

The redox regulatory effect of low doses of O_3_ has been repeatedly described and has led to the inclusion of O_3_ treatments among the strategies to combat mitochondriopathies linked to high oxidative stress and often due to mitochondrial aging and dysfunction [8]. Several studies demonstrated that the O_3_SS-induced response is dependent on the activation of the transduction mechanisms of Nrf2 from cytoplasm to nucleus, inducing antioxidant enzymes synthesis, such as SOD, CAT (catalase), and HO1 (heme oxygenase 1), among others [25,26,27]. In addition, recent preclinical studies demonstrate that O_3_SS mitigates parthanatos after ischemic stroke. In both in vitro (SH-SY5Y cells exposed to H_2_O_2_) and in vivo (murine ischemic stroke) models, O_3_SS administration decreased oxidative stress and neuronal death [28]. Our results confirm the antioxidant activity of O_3_SS.

Intravenous infusion of a 5% glucose solution has made it possible to treat patients who could not be treated with the classical method, achieving similar results [29]. However, O_3_SS allows for precise control of O_3_ dosage and ensures immediate reaction of O_3_ with blood components, leading to the formation of redox-active messengers that trigger antioxidant, anti-inflammatory, and immunomodulatory responses. Additionally, O_3_SS does not introduce an exogenous glucose load, thereby reducing the risk of metabolic complications such as hyperglycemia or electrolyte disturbances, which are documented risks with intravenous glucose infusions, especially in patients with metabolic comorbidities [30,31]. O_3_SS also minimizes the risk of infusion-related complications such as thrombophlebitis, which can occur with repeated glucose infusions [31]. Compared to major autohemotherapy (MAH), which involves ex vivo ozonation of a patient’s blood followed by reinfusion, administration of O_3_SS offers several practical advantages. O_3_SS is technically simpler, avoids direct blood handling, requires a lower caliber needle, and is more acceptable for patients who refuse blood manipulation due to religious or procedural concerns. Additionally, O_3_SS generally carries fewer legal implications since it does not involve extracorporeal blood processing. However, O_3_SS has significant disadvantages as a lack of standardization. In general, the medical literature highlights that, while O_3_SS may be more convenient in certain settings, MAH remains the preferred method for systemic O_3_ therapy due to its superior efficacy, safety, and reproducibility when performed with appropriate protocols and individualized O_3_ concentrations. Notwithstanding, clinical evidence supports the efficacy and safety of both MAH and O_3_SS infusions in diverse medical contexts. In the study by Makarov et al. (2017) [32], elderly patients undergoing rehabilitation for chronic cardiovascular and musculoskeletal conditions received a combination of MAH or O_3_SS and gravitational therapy. Over a follow-up period of up to 7 years, the combination of O_3_SS and gravitational therapy resulted in the most significant reduction in the risk of disease progression and need for surgical intervention, as determined by Cox regression analysis. In contrast, a retrospective analysis conducted by Andryushchenko et al. (2019) [33]. A total of 144 patients aged 17 to 72 years (58% women) hospitalized in an intensive care unit in Kyiv received parenteral O_3_ therapy as an adjunct to standard care for various acute and chronic conditions, including sepsis, trauma, burns, acute infections, diabetes, and peripheral vascular disease. Two main O_3_ therapy modalities were applied: intravenous O_3_SS (200 mL containing 0.48 mg of O_3_) administered across 169 sessions, or MAH, involving 200–400 mL of autologous blood enriched with 1.8 mg of O_3_, across 185 sessions. Clinically, both interventions were associated with improved blood oxygenation, enhanced rheological properties, activation of humoral immunity, and pain relief. Notably, no adverse events or complications were recorded in any of the 304 procedures, underscoring the safety of the approach. The study supports the practical utility of O_3_ therapy in emergency medicine and recommends further standardization and controlled clinical trials to validate and refine its use [13,16].

Microglial cells are considered among the main cellular players in the development of neuroinflammation underlying several chronic neurodegenerative diseases, including Alzheimer’s disease (AD). For this reason, we chose the BV2 in vitro cellular model, analyzing its behavior after exposure to different amounts of O_3_, in the presence or in the absence of LPS. Our results suggest that even with the lowest O_3_ concentrations used, microglia are polarized towards an anti-inflammatory phenotype. O_3_ is able to down-regulate the expression of the pro-inflammatory cytokine *IL-1β*, and the effect is greater after 24 h from treatment, while O_3_ alone is able to induce a greater expression of the anti-inflammatory cytokine *IL-10*, increasing its concentration even after exposure to LPS. Furthermore, the cellular markers of pro-inflammatory polarization (*iNOS*) are decreased, while the anti-inflammatory ones (Arg-1) are increased, as detected by real-time PCR and immunofluorescence.

4-hydroxynonenal (4HNE) obtained from Polyunsaturated fatty acids (PUFA) after O_3_ addition, increases the expression and transactivation activity of Peroxisome proliferator-activated receptor gamma (PPARγ) [34], and PPARγ was shown to reduce oxidative stress in brain tissue, improving mitochondrial function [35], reducing glial inflammation and Amyloid-beta peptide (Aβ) levels in AD transgenic mouse models [36]. Our data confirm the ability of O_3_ to polarize microglial cells towards an anti-inflammatory phenotype, added to cells at low concentrations and dissolved in 0.9% NaCl saline solution. According to the results, 0.9% saline solution exerts a hormetic effect on BV2 microglial cells, characterized by the activation of antioxidant and anti-inflammatory pathways without inducing cytotoxicity at clinically relevant concentrations. The observed upregulation of *Nrf2* and *SOD1* mRNA at 4 and 24 h indicates that O_3_ triggers the Keap1/Nrf2-dependent antioxidant response, leading to increased transcription of genes encoding key antioxidant enzymes such as SOD, which is consistent with the established mechanism of low-dose O_3_ as a eustress inducer [37]. This activation enhances the cellular capacity to ROS, as evidenced by the reduction in ROS levels following LPS stimulation.

The modulation of inflammatory markers further supports the immunoregulatory role of O_3_. Downregulation of *iNOS* mRNA and upregulation of *Arg-1* mRNA reflect a shift from a pro-inflammatory (M1-like) to an anti-inflammatory (M2-like) microglial phenotype, aligning with the literature showing that low-dose O_3_ suppresses pro-inflammatory mediators and promotes anti-inflammatory gene expression [38,39]. This dual action, attenuation of oxidative stress and reprogramming of microglial activation, underpins the therapeutic rationale for O_3_ therapy in neuroinflammatory and neurodegenerative contexts.

## 4. Materials and Methods

### 4.1. Cell Culture and Treatment

The murine microglial cell line BV-2, kindly provided by Dr. Mangino (Sapienza University of Rome, Italy), was cultured in Dulbecco’s Modified Eagle’s Medium (DMEM; Sigma-Aldrich, St. Louis, MO, USA) supplemented with 10% fetal bovine serum (FBS; Sigma-Aldrich, St. Louis, MO, USA) and 1% of penicillin–streptomycin, L-glut and non essential amino acids (Sigma-Aldrich, St. Louis, MO, USA), at 37 °C in a humidified atmosphere containing 5% CO_2_. Human umbilical vein endothelial cells (HUVECs; Lonza, Basel, Switzerland) were cultured in EGM-2 Endothelial Cell Growth Medium BulletKit (Lonza, Basel, Switzerland) at 37 °C in a 5% CO_2_ humidified incubator. Cells were seeded in 6-well plates at a density of 5 × 10^5^ cells/well and incubated at 37 °C.

The human umbilical vein endothelial cells (HUVECs) included in this study were obtained from Lonza (Basel, Switzerland), which collected post-partum umbilical cords with informed consent. As the study did not involve living human subjects and the tissue was obtained as medical waste, Research Ethics Committee approval was not required, in accordance with institutional and national regulations [40,41,42,43,44]. All procedures complied with applicable ethical standards and biosafety guidelines.

The O_3_-saline mixture was prepared using the WeZONO device (Deva Med. Sedecal, Madrid, Spain). Treatments were performed by mixing the O_3_SS with culture medium at a 1:20 dilution. The ozonized saline was prepared by bubbling an O_2_/O_3_ gas mixture (output from the WeZONO device (Deva Med. Sedecal, Madrid, Spain)) set at 1, 5, and 10 µg/mL) into 0.9% saline solution (Eurospital, Trieste, Italy) for 20 min. Based on a previously established and validated methodology [20], the actual concentration of O_3_ dissolved in the saline solution is 10% of the generator output value, resulting in concentrations of 0.1, 0.5, and 1 µg/mL prior to dilution. O_3_ was obtained from clinical-grade oxygen. The accuracy of the O_3_ concentration was guaranteed by an internal Algorithm Calculation Measurement system and did not vary by more than 10% of the selected value, in accordance with ISCO3 quality criteria (International Scientific Committee of Ozone Therapy (ISCO3). Guidelines and Recommendations for Medical Professionals Planning to Acquire a Medical Ozone Generator [45] The equipment used is classified within the European Union as a Class IIb medical device.

### 4.2. Trypan Blue Exclusion Assay

Cell viability was determined by the Trypan-Blue exclusion assay. Trypan-blue (Gibco, Thermo Fisher Scientific, Waltham, MA, USA) exclusion assay is a simple and rapid method measuring cell viability that determines the number of viable cells and dead cells. It is based on the principle that live cells with an intact membrane are able to exclude the dye, whereas dead cells without an intact membrane take up the dye. For the Trypan blue exclusion test, cells were seeded onto 6-well plates at a density of 5 × 10^5^/well. After treatments, cells were detached with 1 × Tripsin-EDTA (Sigma-Aldrich, St. Louis, MO, USA), and 10 μL of cell suspension was mixed with 10 μL of Trypan Blue solution, and cell counts were performed using a Burker chamber (Paul Marienfeld GmbH & Co. KG, Lauda-Königshofen, Germany). Blue-stained cells were considered nonviable [46].

### 4.3. Real-Time Quantitative PRC Analysis

Total RNA was extracted from the control and treated cells using the miRNeasy Micro kit (Qiagen, Hilden, Germany) and quantified using NanoDrop One/OneC (Thermo Fisher Scientific, Waltham, MA, USA). cDNA was generated using the High-Capacity cDNA Reverse Transcription kit (Applied Biosystem, Foster City, CA, USA). Quantitative real-time PCR (qPCR) was performed for each sample in triplicate on an Applied Biosystems 7900HT Fast Real-Time PCR System (Applied Biosystem, Cheshire, UK) through the program SDS2.1.1 (Applied Biosystem, Foster City, CA, USA) using the Power SYBR^®^Green PCR Master Mix (Applied Biosystem, Foster City, CA, USA). The primers for real-time PCR amplification were designed with UCSC GENOME BROWSER (Available online: http://genome.cse.ucsc.edu/ (accessed on 1 November 2022); University of California, Santa Cruz, CA, USA) (Table 1). The primer pair sequences were matched by BLASTn to the genome sequence to identify the primer locations with respect to the exons. A comparative threshold cycle (CT) method was used to analyze the real-time PCR data, where the amount of target, normalized to the endogenous reference of *β-Actin* (∆CT), and relative to the list of primer couples generated for qRT-PCR.

### 4.4. Immunofluorescence

A total of 30,000 BV2 cells/well of the chamber slides were plated in 200 μL of 10% Fetal Bovine Serum (FBS) in DMEM and stimulated with O_3_SS in the presence and absence of lipopolysaccharide (LPS, strain 0111:B4, Sigma-Aldrich, St. Louis, MO, USA) 1 ng/mL for 24 h. After 3 washes in PBS (1x), cells were fixed in 4% paraformaldehyde for 30 min at room temperature. After three washes, cells were permeabilized with 0.1% Triton X-100 for 5 min. Two washes in PBS were performed, and glycine 0.1 M was added for 20 min RT. After two additional washes in PBS, the cells were incubated overnight at 4 °C with the rabbit primary antibody anti-Arginase-1 (AB-84248, Immunological Sciences, Rome, Italy) diluted 1:100 in PBS containing 0.1% BSA [47]. After 3 washes in PBS, the cells were incubated with secondary antibody, goat anti-rabbit IgG (H+L) conjugated with CF488A (Biotium, Fremont, CA, USA, catalog number 20012) for 30 min at room temperature (1:100 in PBS) in the dark. After washing off the excess, the slides were sealed with Vectashield^®^ Antifade Mounting Medium with DAPI (Newark, NJ, USA). Images were acquired and analyzed using NIS-Elements software 4.30.02 (Nikon Instruments Inc., Melville, NY, USA). Using ImageJ software, version 1.48 [48], the fluorescence intensity in the FITC channel was measured for each field and expressed as the mean fluorescence per cell by dividing the total signal by the number of cells measured by DAPI nuclei staining.

### 4.5. Analysis Using the Cell-Permeable Probe 2′,7′-Dichlorodihydrofluorescein Diacetate (H_2_DCFDA)

BV-2 cells were seeded at a density of 50,000 cells per well in chamber slides, in 500 µL of DMEM supplemented with 10% FBS, and subsequently treated with O_3_SS. After 45 min, the cells were stimulated with lipopolysaccharide (LPS) at a final concentration of 1 ng/mL and incubated for an additional 4 h at 37 °C. Following the incubation, cells were washed twice with PBS and fixed in 4% paraformaldehyde for 30 min at room temperature. After fixation, two additional PBS washes were performed. Cells were then incubated with the cell-permeable fluorescent probe H_2_DCFDA at 37 °C for 30 to 45 min. After staining, cells were washed five times with PBS, then incubated with DAPI (1:100 dilution) for 5 min in the dark. Six final PBS washes were carried out. Finally, coverslips were mounted using Vectashield^®^ Antifade Mounting Medium (Newark, NJ, USA). Images were acquired and analyzed using NIS-Elements software (Nikon Instruments Inc., Melville, NY, USA). Fluorescence intensity was quantified with ImageJ software, version 1.48 [49].

### 4.6. Detection of Reactive Chlorine Species (Hypochlorous Acid/Hypochlorite)

The measurement of the hypochlorite anion was performed as a crucial control to rule out that the biological effects observed in our experiments were mediated by the potential formation of hypochlorite (ClO^−^) as a byproduct of O_3_ dissolution, rather than by O_3_ itself. To quantify the levels of hypochlorous acid (HClO), we measured the concentration of the hypochlorite anion (ClO^−^) using a commercial Hypochlorite Detection Kit (Colorimetric) (ab219929, Abcam, Cambridge, UK). This method is based on a sensor that undergoes a specific reaction with ClO^−^ to generate a red-colored product, with absorbance measured at 555 nm. Given that HClO dissociates into H^+^ and ClO^−^ (pKa ≈ 7.5) and the kit detects the anionic form, the results provide an accurate indirect measurement of the total hypochlorous acid/hypochlorite content in the samples. The assay was performed on the cell culture medium mixed with saline solution at the concentrations previously described, using both HUVEC EBM and DMEM from BV2 cell cultures, and results were compared to the unconditioned culture media used as controls.

### 4.7. Statistical Analyses

Data were expressed as the mean values ± standard deviations (SD) or mean values ± SEM from at least three independent experiments. Statistical analyses were performed using the unpaired Student’s *t* test (GraphPad Prism 8.0.1, San Diego, CA, USA). All results were considered statistically significant with *p* < 0.05. Experiments were performed at least in triplicate.

## 5. Conclusions

Given these findings, O_3_SS may represent a promising adjunctive approach in managing conditions characterized by oxidative stress and inflammation, particularly in cases where MAH is not feasible or is contraindicated. Nonetheless, the current lack of methodological standardization and the absence of robust, large-scale randomized controlled trials highlight the need for further investigation. Future research should prioritize well-designed clinical studies to confirm therapeutic efficacy, optimize dosage regimens, and assess long-term safety. In this study, we explored the effects of systemically administered O_3_SS on murine microglial (BV2) and human endothelial (HUVEC) cells, focusing on cytotoxicity, redox regulation, and immunomodulation. Cells were treated with increasing concentrations of O_3_ (1, 5, or 10 μg/NmL) dissolved in saline. Low concentrations (1 and 5 μg/NmL) promoted cell proliferation without inducing cytotoxic effects, whereas the highest concentration (10 μg/NmL) led to decreased viability and increased cell death. O_3_SS enhanced the expression of key antioxidant genes such as *Nrf2* and *SOD1,* and significantly reduced reactive oxygen species in LPS-stimulated microglia. Moreover, O_3_SS shifted the microglial profile toward an anti-inflammatory phenotype, as evidenced by the downregulation of pro-inflammatory markers (*iNOS*, *IL-1β*) and upregulation of anti-inflammatory mediators (*Arg-1*, *IL-10*), confirmed at the protein level via immunofluorescence. Overall, these results support the potential of low-dose O_3_SS to activate endogenous antioxidant defenses and promote immune homeostasis, offering a basis for its further development as a safe and effective systemic O_3_ therapy.

## 6. Limitations

Although our results provide promising evidence of the antioxidant and anti-inflammatory potential of O_3_SS, certain limitations must be acknowledged. First, the study was conducted exclusively on cell cultures in vitro (murine microglia BV2 and human endothelial cells HUVEC), which, although informative, do not fully replicate the complexity of physiological environments in vivo. Furthermore, the effects of repeated or chronic exposure to O_3_SS were not evaluated, and the short-term cellular responses observed here may not reflect long-term outcomes. Finally, although we observed modulation of antioxidant and inflammatory markers, the molecular mechanisms underlying these effects, such as specific signaling pathways or interactions with redox-sensitive transcription factors, require further investigation. Clinically, O_3_ therapy (via O_3_SS or MAH) has shown efficacy, without serious adverse effects reported. While both methods have demonstrated safety and potential efficacy, current limitations include variability in O_3_ dosing, delivery protocols, and a lack of universal guidelines. Future in vivo studies are essential to validate these findings, determine pharmacokinetics, and establish safe and effective dosing protocols for potential clinical applications.

## Figures and Tables

**Figure 1 molecules-30-03932-f001:**
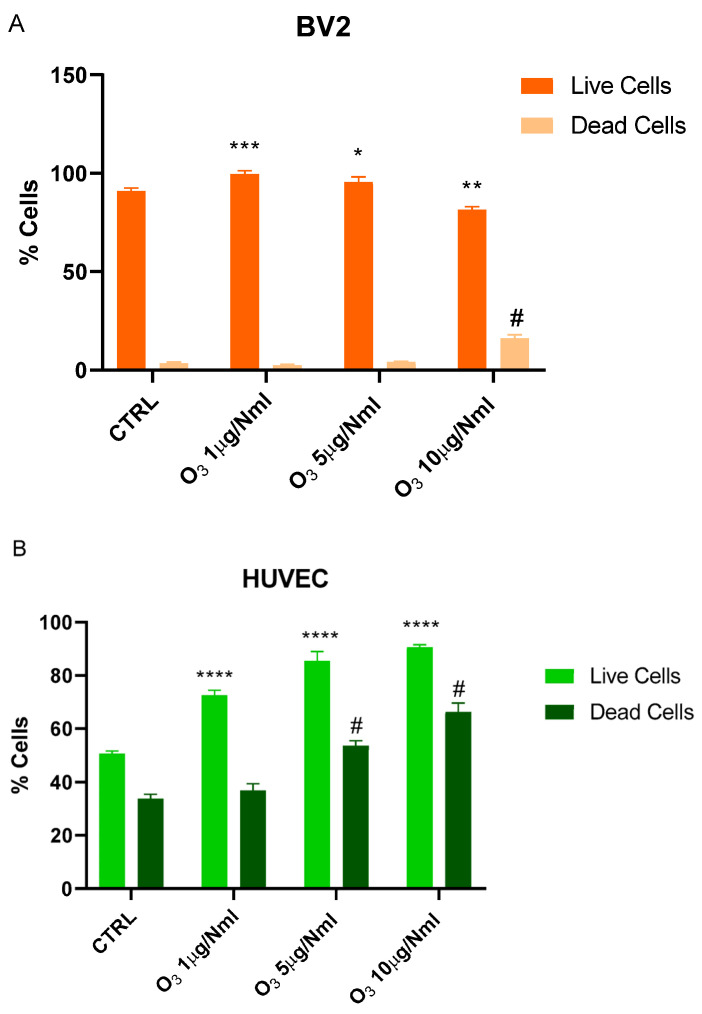
Trypan Blue Assay for O_3_ cytotoxicity. (**A**) BV2 cells were treated with 1, 5, or 10 μg/NmL of O_3_. (**B**) HUVEC cells were treated with 1, 5, or 10 μg/NmL of O_3_. Data are reported as mean ± SD from three independent experiments; statistical analysis was performed using the unpaired Student’s *t*-test. * Vs. Live cells; # Vs. Dead cells; * *p* < 0.05; ** *p* < 0.01; *** *p* < 0.001; **** *p* < 0.0001; # *p* < 0.05. CTRL (control).

**Figure 2 molecules-30-03932-f002:**
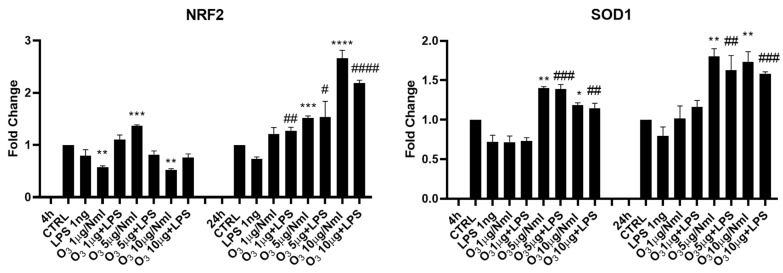
mRNA expression of *Nrf2* and *SOD1* was evaluated by qRT-PCR. Data are shown as mean ± SD from three independent experiments performed in triplicate. Expression profiles were determined using the 2^−ΔΔCT^ method. Statistical analysis was reported using unpaired Student’s *t*-test. * Vs. CTRL; # Vs. LPS; * *p* < 0.05; ** *p* < 0.01; *** *p* < 0.001; **** *p* < 0.0001; # *p* < 0.05; ## *p* < 0.01; ### *p* < 0.001; #### *p* < 0.0001. CTRL (control) LPS (Lipopolysaccharide).

**Figure 3 molecules-30-03932-f003:**
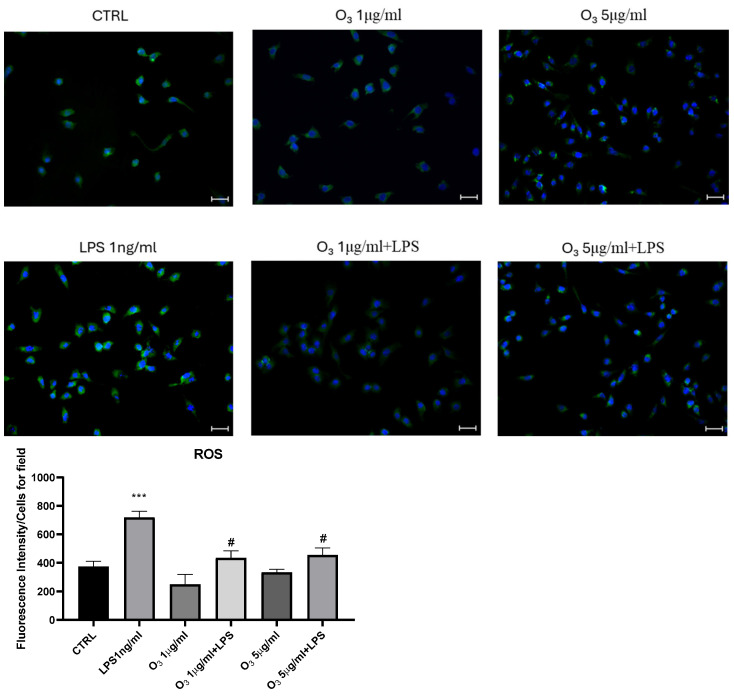
O_3_ treatment reduces ROS expression in the LPS-stimulated BV2 cells. Analysis using the cell-permeable probe 2′,7′-dichlorodihydrofluorescein diacetate (H_2_DCFDA). Quantification of the median fluorescence of the cell-permeable probe 2′,7′-dichlorodihydrofluorescein diacetate using ImageJ version 1.48. The data are expressed as histograms, normalized to the number of cells in field. 4′,6-diamidino-2-phenylindole (DAPI) was used to counterstain the nuclei. Vs. CTRL *** *p* < 0.001; Vs. LPS # *p* < 0.05; magnification: 20×.

**Figure 4 molecules-30-03932-f004:**
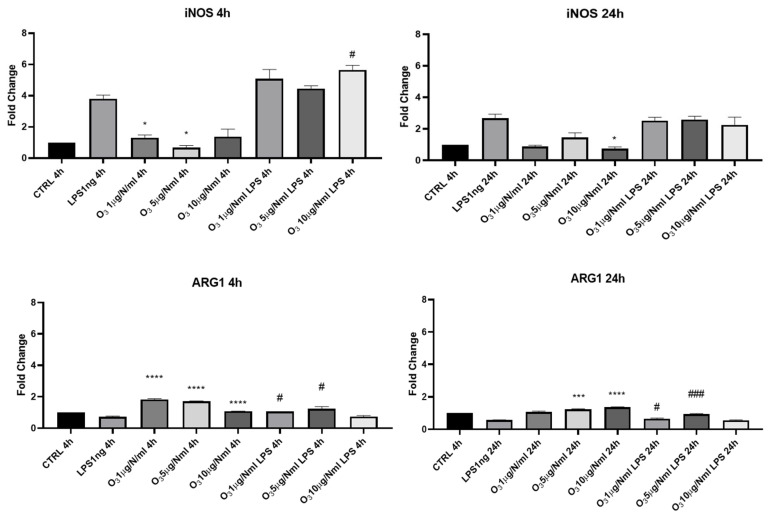
mRNA expression of *iNOS* and *ARG1* was evaluated by qRT-PCR. Data are shown as mean ± SD from three independent experiments performed in triplicate. Expression profiles were determined using the 2^−ΔΔCT^ method. Statistical analysis was reported using unpaired Student’s *t*-test. * Vs. CTRL; # Vs. LPS; * *p* < 0.05; *** *p* < 0.001; **** *p* < 0.0001; # *p* < 0.05; ### *p* < 0.001.

**Figure 5 molecules-30-03932-f005:**
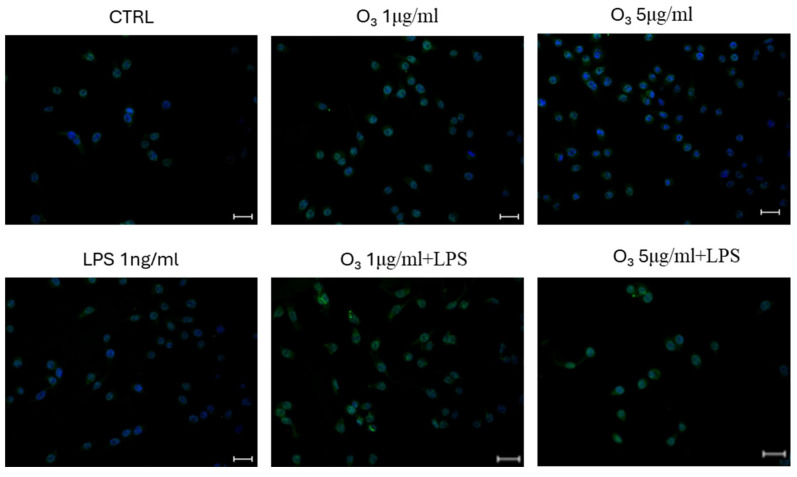
Immunofluorescence analysis of ARG-1 expression in BV2 cells cultured in the presence or absence of LPS for 24 h. Quantification of the median fluorescence intensity was performed by ImageJ software, and data were expressed as histograms, normalized to the number of cells in field. 4′,6-diamidino-2-phenylindole (DAPI) was used to counterstain the nuclei. Data are expressed as mean ± SD for each group (n = 3). Statistical analysis was performed by an unpaired Student’s *t*-test. Vs. CTRL ** *p* < 0.01; Vs. LPS ### *p* < 0.001; magnification: 20×.

**Figure 6 molecules-30-03932-f006:**
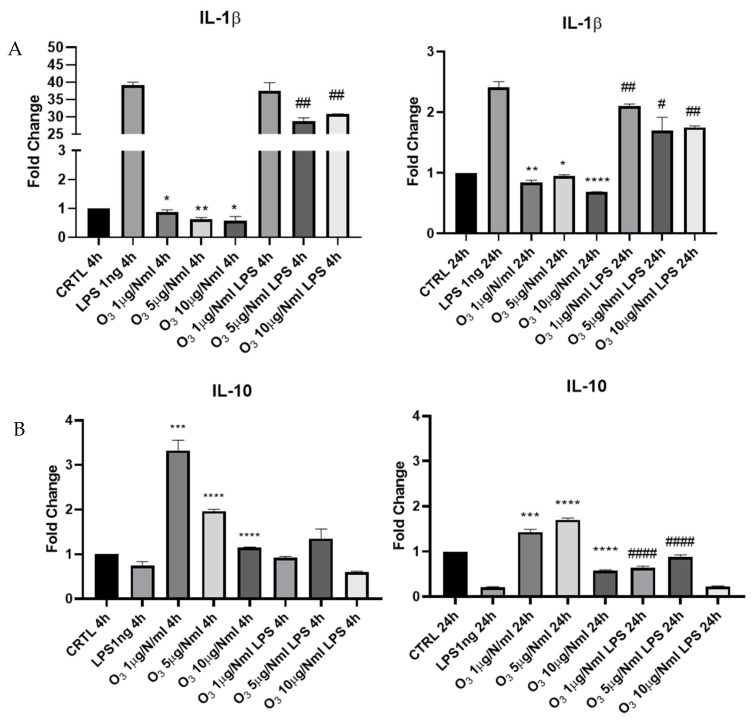
Evaluation of mRNA expression of *IL-1β* (**A**) and *IL-10* (**B**) assessed by qRT-PCR. Data are shown as mean ± SD from three independent experiments performed in triplicate. Expression profiles were determined using the 2^−ΔΔCT^ method. Statistical analysis was reported using unpaired Student’s *t*-test. Vs. CTRL * *p* < 0.05; ** *p* < 0.01; *** *p* < 0.001; **** *p* < 0.0001; Vs. LPS # *p* < 0.05; ## *p* < 0.01; #### *p* < 0.0001.

**Table 1 molecules-30-03932-t001:** Primers used for real-time PCR amplification.

Gene	Forward Primer (5′–3′)	Reverse Primer (5′–3′)	Accession Numbers
*mIL-1β*	GAAATGCCACCTTTTGACAGTG	TGGATGCTCTCATCAGGACAG	NM_008361.4
*mIL-10*	GCCCTTTGCTATGGTGTCCTTTC	TCCCTGGTTTCTCTTCCCAAGAC	NM_010548.2
*mARG1*	ATGTGCCCTCTGTCTTTTAGGG	GGTCTCTCACGTCATACTCTGT	NM_007482.3
*miNOS*	GGCAGCCTGTGAGACCTTTG	GCATTGGAAGTGAAGCGTTTC	AF427516.1
*mACT-β*	GGCTGTATTCCCCTCCATCG	CCAGTTGGTAACAATGCCATGT	NM_007393.5
*mNRF2*	TCTGAGCCAGGACTACGACG	GAGGTGGTGGTGGTGTCTCTGC	NM_010902
*mSOD1*	GCCCGCTAAGTGCTGAGTC	AGCCCCAGAAGGATAACGGA	NM_017050

## Data Availability

The original contributions presented in this study are included in the article.

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
