# Peer review of "Ozone Saline Solution Polarizes Microglial Cells Towards an Anti-Inflammatory Phenotype"

_molecules, 2025, doi:10.3390/molecules30193932_

Round 1

Reviewer 1 Report

Comments and Suggestions for Authors

The authors evaluate the cytotoxicity, antioxidant response, and immunomodulatory effects of O3SS in two cell cultures.

A minor revision is needed. The comments are below:

- Add a section/topic explaining the unnecessary study approval number by a Research Ethics Committee to use HUVEC cells, if was in vitro.

- Why is the 10 µg/mL group (with or without LPS) not shown or investigated in Figures 3 and 5?

- Provide the type of ozone generator, manufacturer details, whether it was medical-grade, and a calibration statement.

- Inform the ozone feed rate and/or ozone dosage applied (see DOI: 10.1080/01919512.2015.1006467 ).

- The three ozone concentrations chosen and the 20-minute ozonation based on any previous reference or study? Such as the Madrid Declaration? Clarified.

- How did ozone diffusion occur in NaCl? What is the bubble diffuser? Provide more details about this and the ozone mass transfer process.

- How was the O3 concentration in NaCl measured? Explain that.

- Was hydrogen peroxide formation measured? If not, comment.

- Traces of Fe++ were found in NaCl? Could it catalyze OH- radicals (see the Fenton’s reaction). Discuss.

- The pH was measured in the various medium/solution conditions (before and after the O3SS)?

- Some studies used ascorbate (vitamin C) saline solution to compare the O3SS. Did the authors consider this? Discuss.

- Discuss your results with the work of Ikonomidis et al. (2005), which corroborates your findings (ozone concentrations toxicity in the O3SS), but with a different approach. (See Riv Ital. Ossigeno-Ozonoterapia; 4:40-43).

- Despite mentioning MAH, why didn't the authors compare O3SS results to blood ozonation? This is the most established modality.

- Reference #10 is inaccessible. Please provide a link.

- Reviewed the reference #31. Title, doi…

Author Response

The authors evaluate the cytotoxicity, antioxidant response, and immunomodulatory effects of O3SS in two cell cultures.

A minor revision is needed. The comments are below:

- Add a section/topic explaining the unnecessary study approval number by a Research Ethics Committee to use HUVEC cells, if was in vitro.

Research Ethics Committee (REC) approval is not required for this study because it exclusively involves the use of commercial (Lonza, Basel, Switzerland) human umbilical vein endothelial cells (HUVECs) isolated from postpartum umbilical cords for in vitro experimentation. The collection of umbilical cords is a noninvasive procedure performed after delivery, and the tissue is considered medical waste. Informed consent for the use of the tissue is routinely obtained, which addresses the primary ethical consideration for such research.[1-2][5-6]

According to current consensus in the medical literature, the isolation and culture of HUVECs for in vitro studies does not involve living human subjects, nor does it pose direct risks to donors, and is therefore generally exempt from mandatory REC approval in most jurisdictions.[1-6] This position is supported by established protocols and reviews, which consistently describe HUVEC-based research as ethically straightforward when proper consent is obtained and biosafety measures are followed.[1-6] Nevertheless, all procedures were conducted in accordance with institutional guidelines and relevant national regulations. 

In section 4, page 11, line 319 we included:

The human umbilical vein endothelial cells (HUVECs) included in this study were obtained from Lonza (Basel, Switzerland) that collected post-partum umbilical cords with informed consent. As the study did not involve living human subjects and the tissue was obtained as medical waste, Research Ethics Committee approval was not required, in accordance with institutional and national regulations.[1-6] All procedures complied with applicable ethical standards and biosafety guidelines.

  1. Isolation and Culture of Human Umbilical Vein Endothelial Cells (HUVECs). Chandel S, Kumaragurubaran R, Giri H, Dixit M. Methods in Molecular Biology (Clifton, N.J.). 2024;2711:147-162. doi:10.1007/978-1-0716-3429-5_12.

  1. Culture of Human Endothelial Cells From Umbilical Veins. Siow RC. Methods in Molecular Biology (Clifton, N.J.). 2012;806:265-74. doi:10.1007/978-1-61779-367-7_18.

  1. Human Umbilical Vein Endothelial Cells (HUVECs) in Pharmacology and Toxicology: A Review. Cao Y. Journal of Applied Toxicology : JAT. 2025;. doi:10.1002/jat.4885.

  1. A New, Rapid and Reproducible Method to Obtain High Quality Endothelium in Vitro. Jiménez N, Krouwer VJ, Post JA. Cytotechnology. 2013;65(1):1-14. doi:10.1007/s10616-012-9459-9.

  1. A Simple and Biosafe Method for Isolation of Human Umbilical Vein Endothelial Cells. Lei J, Peng S, Samuel SB, et al. Analytical Biochemistry. 2016;508:15-8. doi:10.1016/j.ab.2016.06.018.

  1. ESTIV Questionnaire on the Acquisition and Use of Primary Human Cells and Tissue in Toxicology. Sladowski D, Combes R, van der Valk J, Nawrot I, Gut G. Toxicology in Vitro : An International Journal Published in Association With BIBRA. 2005;19(7):1009-13. doi:10.1016/j.tiv.2005.06.041.

- Why is the 10 µg/mL group (with or without LPS) not shown or investigated in Figures 3 and 5?

The optimal doses of ozone to be added to the cells were determined from cytotoxicity curves and real-time PCR experiments designed to quantify the synthesis of mRNAs for pro- and anti-inflammatory cytokines and for markers of pro- or anti-inflammatory polarization of BV2 microglial cells. The results of these experiments, reported in Figs. 1, 4, and 6, respectively, indicated that the dose of 5 µg/mL gave the best results. Consequently, this concentration was used in subsequent experiments, reported in Figs. 3 and 5.

- Provide the type of ozone generator, manufacturer details, whether it was medical-grade, and a calibration statement.

- Inform the ozone feed rate and/or ozone dosage applied (see DOI: 10.1080/01919512.2015.1006467 ).

In Page 11, line 319 The ozone-saline mixture was prepared using the WeZONO device (Deva Med. Italy).

Authors add:

The ozone-saline mixture was prepared using the WeZONO device (Deva Med. Italy). Ozone was obtained from clinical-grade oxygen. The accuracy of the ozone concentration was guaranteed by an internal Algorithm Calculation Measurement system and did not vary by more than 10% of the selected value, in accordance with ISCO3 quality criteria (1). The equipment used is classified within the European Union as a Class IIb medical device.

Reference:

  1. International Scientific Committee of Ozone Therapy (ISCO3). (2019). Guidelines and Recommendations for Medical Professionals Planning to Acquire a Medical Ozone Generator. ISCO3. Retrieved from https://isco3.org/wp-content/uploads/2015/09/Generadores-ISCO3-SOP-DEV-01-01-2019.pdf

- The three ozone concentrations chosen and the 20-minute ozonation based on any previous reference or study? Such as the Madrid Declaration? Clarified.

- How did ozone diffusion occur in NaCl? What is the bubble diffuser? Provide more details about this and the ozone mass transfer process.

Page 11, line 316 -318

Ozone gas (O₃) was added to the wells after being dissolved in 0.9% saline solution (Eurospital, Trieste, Italy) by bubbling for 20 minutes at final concentrations of 1, 5, or 10 µg/NmL.

Authors add:

Ozone gas (O₃) was added to the wells after being dissolved in 0.9% saline solution (Eurospital, Trieste, Italy) by bubbling the solution for 20 minutes using the medical devise Bexozone (Bexen Medical, Guipuzkoa, Spain) and a 1.2 mm needle, at final concentrations of 1, 5, or 10 µg/NmL. The O₃ concentration and the bubbling time were selected to simulate the clinical use of Ozonized Saline Solution. The ozone-saline mixture was prepared using the WeZONO device (Deva Med. Italy). Treatments were performed by mixing the ozonized saline solution with culture medium at a 1:20 dilution. The ozonized saline was prepared by bubbling an O₂/O₃ gas mixture (output from the WeZONO device (Deva Med. Italy) set at 1, 5, and 10 µg/mL) into 0.9% saline solution (Eurospital, Trieste, Italy) for 20 minutes. Based on a previously established and validated methodology [20], the actual concentration of ozone dissolved in the saline solution is 10% of the generator output value, resulting in concentrations of 0.1, 0.5, and 1 µg/mL prior to dilution.

Reference:

Schwartz Tapia, A. ISCO3/MET/00/21 Ozonized Saline Solution (O3SS). 2025.

- How was the O3 concentration in NaCl measured? Explain that.

- Was hydrogen peroxide formation measured? If not, comment.

On Page 8 line 182-183

Moreover, it was demonstrated that ozonation of the saline solution eliminates traces of Bromine that exist in the normal pharmacological formulation of the saline [19].

Authors add:

Moreover, it was demonstrated that ozonation of the saline solution eliminates traces of Bromine that exist in the normal pharmacological formulation of the saline [19]. The kinetics of ozone saturation in the saline solution were previously studied using spectrophotometric methods. At the concentrations used and with the same medical device and conditions, the ozone concentration in the solution stabilizes after 10 minutes of bubbling and corresponds to 10% of the initial concentration. Under these conditions, the formation of hypochlorous acid or hydrogen peroxide was not detected (1).

  1. Martínez Sánchez, Gregorio. (2020). Practical aspects in ozone therapy: Study of the ozone concentration in the ozonized saline solution, Ozone Therapy Global Journal, 10, nº 1, pp 55-68

- Traces of Fe++ were found in NaCl? Could it catalyze OH- radicals (see the Fenton’s reaction). Discuss.

The presence of traces of ferrous ions (Fe²⁺) in sodium chloride (NaCl) can catalyze the formation of hydroxyl radicals (•OH) via the Fenton reaction, but only if hydrogen peroxide (H₂O₂) or another suitable peroxide is present. The classical Fenton reaction is:

Fe²⁺ + H₂O₂ → Fe³⁺ + •OH + OH⁻

In pure NaCl solution, without added peroxide, this reaction does not occur. However, if H₂O₂ is introduced, Fe²⁺ can efficiently catalyze •OH generation, as demonstrated in multiple studies of Fenton and Fenton-like systems.

The saline solution used was of pharmaceutical quality and met the characteristics established by the European Pharmacopoeia. In the case of iron, a maximum content of 2 ppm is established. For the Fenton reaction to take place, hydrogen peroxide must be present in the medium. Several studies have shown that under the conditions in which the solution was ozonated (concentration values between 1 and 10 µg/mL and 0.9% sodium chloride), hydrogen peroxide is not formed. Therefore, the formation of the hydroxyl radical is ruled out.

The scientific literature indicates that, under ozonation conditions in 0.9% sodium chloride solution and with ozone concentrations between 1 and 10 µg/mL, hydrogen peroxide is not formed in detectable amounts, and therefore the formation of hydroxyl radicals (•OH) is ruled out in the absence of organic precursors or additional catalysts. Studies on ozonation in simple aqueous matrices, such as saline solution, demonstrate that H₂O₂ generation requires the presence of reactive organic compounds or specific conditions like cavitation, which are not present in standard ozonation of NaCl at neutral pH (1-3).

Analysis of H2O2 in samples of 0.9% HOCl done by analytical chemistry methods did not reveal any accumulation of H2O2 in concentrations exceeding 0.002 % in any of the ozonation schemes, although it was found even much lower concentrations, in the order of 0.0004 % (4).

Reference:

  1. Ozonation of Drinking Water: Part I. Oxidation Kinetics and Product Formation. von Gunten U. Water Research. 2003;37(7):1443-67. doi:10.1016/S0043-1354(02)00457-8.

  1. The Basics of Oxidants in Water Treatment. Part B: Ozone Reactions. von Gunten U. Water Science and Technology : A Journal of the International Association on Water Pollution Research. 2007;55(12):25-9. doi:10.2166/wst.2007.382.

  1. Towards Reducing DBP Formation Potential of Drinking Water by Favouring Direct Ozone Over Hydroxyl Radical Reactions During Ozonation. De Vera GA, Stalter D, Gernjak W, et al. Water Research. 2015;87:49-58. doi:10.1016/j.watres.2015.09.007.

  1. Maslennikov OV, Kontorshikova CN, Gribkova IA. Ozone therapy in Practice. Health Manual, Ministry Health Service of The Russian Federation The State Medical Academy Of Nizhny Novgorod, Russia. http://www.absoluteozone.com/assets/ozone_therapy_in_practice.pdf. 1 ed2008.

In the manuscript Page 8 line 182-183

Authors add:

Traces of Fe²⁺ in sodium chloride can, in principle, catalyze hydroxyl radical (•OH) generation through the Fenton reaction. However, this requires the presence of H₂O₂ or another suitable peroxide. In pure pharmaceutical-grade NaCl solution (0.9% saline), the European Pharmacopoeia sets a maximum Fe content of 2 ppm, and under standard ozonation conditions (1–10 µg/mL O₃), no detectable H₂O₂ is formed. Analytical studies confirm H₂O₂ levels remain far below the threshold necessary for Fenton chemistry. Therefore, while Fe²⁺ traces could act as catalysts in theory, in practice under these conditions the formation of hydroxyl radicals via the Fenton reaction is ruled out (1).

References:

  1. Towards Reducing DBP Formation Potential of Drinking Water by Favouring Direct Ozone Over Hydroxyl Radical Reactions During Ozonation. De Vera GA, Stalter D, Gernjak W, et al. Water Research. 2015;87:49-58. doi:10.1016/j.watres.2015.09.007.

- The pH was measured in the various medium/solution conditions (before and after the O3SS)?

Significant pH variations are unlikely when O₃SS is added to Dulbecco’s Modified Eagle’s Medium (DMEM) at a 1:20 ratio, due to the buffering capacity of DMEM. However, it is important to note that DMEM’s bicarbonate-based buffer can be susceptible to pH drift if exposed to room air for prolonged periods, as loss of CO₂ may cause a gradual increase in pH, but this effect is not directly related to the addition of O₃SS at the specified ratio and conditions of the experiment.

Reference:

pH Drift of "Physiological Buffers" and Culture Media Used for Cell Incubation During in Vitro Studies. Lelong IH, Rebel G. Journal of Pharmacological and Toxicological Methods. 1998;39(4):203-10. doi:10.1016/s1056-8719(98)00019-7.

In the manuscript Page 8 line 182-183

Authors add:

The proportion of O₃SS added to the DMEM medium was 1:20. Since DMEM contains a buffering system, significant pH variations after the addition of O₃SS were not expected.

- Some studies used ascorbate (vitamin C) saline solution to compare the O3SS. Did the authors consider this? Discuss.

A direct comparative study between ascorbate (vitamin C) saline solution and ozonized saline solution (O₃SS) was not the purpose of this research. The authors did not design the study to evaluate or contrast the cellular or clinical effects of ascorbate saline versus O₃SS.

The medical literature demonstrates that ascorbate saline and O₃SS have fundamentally different mechanisms and profiles. Ascorbate saline can induce cytotoxicity in vitro due to hydrogen peroxide (H₂O₂) generation, especially in the presence of transition metals, and is used clinically for short-term treatment of scurvy and as an investigational agent in oncology and critical illness.[1] O₃SS, in contrast, has been studied for its antioxidant, anti-inflammatory, and immunomodulatory effects, with evidence of safety and efficacy in clinical and in vitro settings, and does not induce significant cytotoxicity at therapeutic concentrations.[2-7]

Some studies have explored the interaction between ascorbate and ozone, showing that their combination can produce cytotoxic ozonides and free radicals under acidic conditions, but these findings are context-specific and not directly relevant to the current study design.[7] Therefore, while the literature supports the importance of distinguishing between these agents, the authors did not consider a direct comparison with ascorbate saline solution in this research.

Reference:

  1. Administration of Intravenous Ascorbic Acid-Practical Considerations for Clinicians. Walker SE, Iazzetta J, Law S, et al. Nutrients. 2019;11(9):E1994. doi:10.3390/nu11091994.

  1. A Pilot Study for Treatment of COVID-19 Patients in Moderate Stage Using Intravenous Administration of Ozonized Saline as an Adjuvant Treatment-Registered Clinical Trial. Sharma A, Shah M, Lakshmi S, et al.
  2. International Immunopharmacology. 2021;96:107743. doi:10.1016/j.intimp.2021.107743.

  1. Comparison of Microbubbling and Conventional Bubbling Methods for Ozonated Saline Solution in CKD Patients: A Pilot Study. Guevara-Aguilar E, Moroni-González D, Jiménez-Ortega JC, Treviño S, Sarmiento-Ortega VE. Free Radical Research. 2025 Mar-Apr;59(4):297-307. doi:10.1080/10715762.2025.2483454.

  1. Ozone Mediators Effect on "In Vitro" Scratch Wound Closure. Valacchi G, Sticozzi C, Zanardi I, et al. Free Radical Research. 2016;50(9):1022-31. doi:10.1080/10715762.2016.1219731.

  1. Intravenous Ozonized Saline Therapy as Prophylaxis for Healthcare Workers (HCWs) in a Dedicated COVID-19 Hospital in India - A Retrospective Study. Sharma A, Shah M, Sane H, et al. European Review for Medical and Pharmacological Sciences. 2021;25(9):3632-3639. doi:10.26355/eurrev_202105_25847.

  1. Acidity Enhances the Formation of a Persistent Ozonide at Aqueous Ascorbate/­Ozone Gas Interfaces. Enami S, Hoffmann MR, Colussi AJ. Proceedings of the National Academy of Sciences of the United States of America. 2008;105(21):7365-9. doi:10.1073/pnas.0710791105.

- Discuss your results with the work of Ikonomidis et al. (2005), which corroborates your findings (ozone concentrations toxicity in the O3SS), but with a different approach. (See Riv Ital. Ossigeno-Ozonoterapia; 4:40-43).

The work by Ikonomidis et al. (2005) addresses general aspects of the clinical application of ozonated saline solution as practiced by the Russian school of ozone therapy. However, it does not provide experimental data that could be directly related to our study. For this reason, we did not consider it appropriate to include a detailed discussion of that work in relation to our results.

Reference:

  1. Ikonomidis, P. Tsaousis, A. Fyntanis, Em. Iliakis. New Data regarding the Use of Oxidative Stress (Ozone Therapy) in the Former Soviet Union Countries. Rivista Italiana di Ossigeno-Ozonoterapia 4: 40-43, 2005.

- Despite mentioning MAH, why didn't the authors compare O3SS results to blood ozonation? This is the most established modality.

Indeed, the main systemic administration routes of ozone are major autohemotherapy (MAH) and ozonated saline solution (O₃SS). Designing an ad hoc experiment to directly compare the effects of these two procedures in cellular models would certainly be of great interest. However, the scope of the present study did not include such a comparison and was specifically focused on evaluating the effects of O₃SS. Nevertheless, we agree with the reviewer’s suggestion and will include this point among the recommendations for future investigations.

Page 11 line 303 we add:

Future studies should be designed to directly compare the effects of O3SS with MAH in cellular models, in order to better delineate the specific contributions of each systemic administration route.

- Reference #10 is inaccessible. Please provide a link.

Ref 10: Schwartz Tapia, A. ISCO3/MET/00/21 Ozonized Saline Solution (O3SS). 2025.

Was replaced by

Schwartz Tapia, A. ISCO3/MET/00/21 Ozonized Saline Solution (O3SS). 2025. Available online: https://isco3.org/officialdocs/

- Reviewed the reference #31. Title, doi…

- Reviewed the reference #31. Title, doi…

Andryushchenko, V.V. Current Issues of Practical Application of Parenteral Ozone Therapy in Emergency Medicine. Emergency Medicine 2019, 8, 121-127.

Authors Reply:

Was replaced by

Andryushchenko, V.V.; Kurdil, N.V.; Struk, V.F.; Kalish, M.M. Actual Issues of the Practical Use of Parenteral Ozone Therapy in Emergency Medicine. Emergency Medicine 2019, 8, 121–127. https://doi.org/10.22141/2224-0586.8.103.2019.192383

Reviewer 2 Report

Comments and Suggestions for Authors

The authors investigated the cytotoxicity, antioxidant response, and immunomodulatory effects of ozonated saline solution using murine microglial and human endothelial cells, indicating a shift toward an anti-inflammatory state. The sound methodology was used and results are correctly presented. This study may be significant for further investigations of potential systemic ozone therapy. Manuscript can be accepted for publication after some corrections listed below:

1) The number of keywords should be reduced and adjusted to the scope of the article.

2) The first sentence in the section Introduction is too long and difficult to read and understand. It would be useful to divide and clarify the above description of ozone, its origin and effects, into several sentences with appropriate citations.

3) In several places in the text (lines 73-77 and 174-176), the labels i), ii), are used for numbering, which is a bit confusing. Perhaps the clarity would be increased if the entire text were not in the same paragraph or without the listed labels.

4) The section Conclusions is too short, thus the significance of obtained results should be more highlighted in terms of “Given these findings,”.

5) In the subsection “4.6 Hypochlorite Detection”, hypochlorite and HClO, hypochlorous acid, are not the same compounds. Hypochlorite is an oxyanion with the chemical formula ClO, which combines with a number of cations to form hypochlorite salts.

Author Response

The authors investigated the cytotoxicity, antioxidant response, and immunomodulatory effects of ozonated saline solution using murine microglial and human endothelial cells, indicating a shift toward an anti-inflammatory state. The sound methodology was used and results are correctly presented. This study may be significant for further investigations of potential systemic ozone therapy. Manuscript can be accepted for publication after some corrections listed below:

  • The number of keywords should be reduced and adjusted to the scope of the article.

We thank the referee for the suggestion. We have reduced the number of keywords and modified as follows: Ozone therapy; Ozonated saline solution; Microglia; Anti-inflammatory response; Nrf2; Reactive oxygen species; Cytokines; Immuno-modulation.

2) The first sentence in the section Introduction is too long and difficult to read and understand. It would be useful to divide and clarify the above description of ozone, its origin and effects, into several sentences with appropriate citations.

Thank you, we have improved the sentence according to your suggestions:

Ozone molecules, formed by three oxygen atoms, are highly unstable. They can quickly break down into one O₂ molecule and one single oxygen atom, which acts as a strong oxidant. Ozone is formed in the atmosphere as a result of electrical discharges and UV light during thunderstorms [1]. In the stratosphere (between 15 and 35 km above sea level), its function is protective, as it shields the Earth from harmful UV rays. In contrast, at the level of the troposphere (10–15 km from the ground), ozone is produced due to the action of nitrogen oxides and volatile organic pollutants. Together with these pollutants, ozone causes damage to the respiratory system, predisposing individuals to the development of even serious pathologies [1].

3) In several places in the text (lines 73-77 and 174-176), the labels i), ii), are used for numbering, which is a bit confusing. Perhaps the clarity would be increased if the entire text were not in the same paragraph or without the listed labels.

We used the labels i) and ii) according to the american custom; if you prefer we can use a) and b) or 1) and 2). In any case we transformed the sentence as follows:

We took advantage of primary cultures of endothelial cells to analyze a possible toxic effect that in vivo could generate phlebitis.  We have used also a mouse model of microglia, as a great interest is focusing on the possible use of ozone in the treatment of neurodegenerative diseases such as Alzheimer's and Parkinson's

We aimed at analyzing two important issues: a putative toxicity of O3SS and the antioxidant and anti-inflammatory activity of the different amounts of ozone dissolved in the saline solution.

4) The section Conclusions is too short, thus the significance of obtained results should be more highlighted in terms of “Given these findings,”.

Thank you, we have changed the conclusions according to your suggestions..

Given these findings, O₃SS may represent a promising adjunctive approach in managing conditions characterized by oxidative stress and inflammation, particularly in cases where MAH is not feasible or is contraindicated. Nonetheless, the current lack of methodological standardization and the absence of robust, large-scale randomized controlled trials highlight the need for further investigations. Future research should prioritize well-designed clinical studies to confirm therapeutic efficacy, optimize dosage regimens, and assess long-term safety. In this study, we explored the effects of systemically administered O₃SS on murine microglial (BV2) and human endothelial (HUVEC) cells, focusing on cytotoxicity, redox regulation, and immunomodulation. Cells were treated with increasing concentrations of ozone (1, 5, or 10 μg/NmL) dissolved in saline. Low concentrations (1 and 5 μg/NmL) promoted cell proliferation without inducing cytotoxic effects, whereas the highest concentration (10 μg/NmL) led to decreased viability and increased cell death. O₃SS enhanced the expression of key antioxidant genes such as Nrf2 and SOD1, and significantly reduced reactive oxygen species in LPS-stimulated microglia. Moreover, O₃SS shifted the microglial profile toward an anti-inflammatory phenotype, as evidenced by the downregulation of pro-inflammatory markers (iNOS, IL-1β) and upregulation of anti-inflammatory mediators (Arg-1, IL-10), confirmed at the protein level via immuno-fluorescence. Overall, these results support the potential of low-dose O₃SS to activate endogenous antioxidant defenses and promote immune homeostasis, offering a basis for its further development as a safe and effective systemic ozone therapy.

5) In the subsection “4.6 Hypochlorite Detection”, hypochlorite and HClO, hypochlorous acid, are not the same compounds. Hypochlorite is an oxyanion with the chemical formula ClO, which combines with a number of cations to form hypochlorite salts.

Response: We sincerely thank the reviewer for this crucial and accurate observation. We apologize for the error in equating hypochlorite (ClO⁻) with hypochlorous acid (HClO). The reviewer is absolutely correct; they are distinct chemical species in a pH-dependent equilibrium.

The commercial detection kit we used (ab219929, Abcam) is indeed designed to detect the hypochlorite anion (ClO⁻) colorimetrically. However, in the context of a biological system (like our study), the measured ClO⁻ levels are directly used to infer the total concentration of reactive chlorine species (RCS), which is the sum of HClO and ClO⁻. This is a standard practice, as the kit's sensitivity allows for the indirect quantification of the HClO/ClO⁻ pool.

We have revised the manuscript text to clearly state that the kit detects the anion and to explain the rationale for its use in measuring hypochlorous acid levels indirectly. The modification aims to eliminate the chemical inaccuracy and improve the methodological description. Thank you again for catching this important oversight.

Text on  (Page 14, lines 381-384)

..4.6 Hypochlorite Detection
The concentration of hypochlorite (HClO, hypochlorous acid) was measured using the Hypochlorite Detection Kit (Colorimetric) (ab219929, Abcam, Cambridge, UK), which is based on a selective sensor that specifically reacts with hypochlorite to produce a red-colored product detectable by absorbance at 555 nm…

Was modified as

  • Detection of Reactive Chlorine Species (Hypochlorous Acid/Hypochlorite)

This measurement was performed as a crucial control to rule out that the biological effects observed in our experiments were mediated by the potential formation of hypochlorite (ClO⁻) as a byproduct of ozone dissolution, rather than by ozone itself. To quantify the levels of hypochlorous acid (HClO), we measured the concentration of the hypochlorite anion (ClO⁻) using a commercial Hypochlorite Detection Kit (Colorimetric) (ab219929, Abcam, Cambridge, UK). This method is based on a sensor that undergoes a specific reaction with ClO⁻ to generate a red-colored product, with absorbance measured at 555 nm. Given that HClO dissociates into H⁺ and ClO⁻ (pKa ≈ 7.5) and the kit detects the anionic form, the results provide an accurate indirect measurement of the total hypochlorous acid/hypochlorite content in the samples.

Reviewer 3 Report

Comments and Suggestions for Authors

The research topic is undoubtedly interesting and potentially valuable. However, the manuscript appears to have been prepared somewhat hastily, which may give the impression that the authors are relying on the peer review process to refine and improve the text. It is essential that manuscripts are clearly written and well-structured prior to submission, allowing reviewers to focus on evaluating the scientific merit rather than addressing issues related to presentation and clarity.

Furthermore, the aim of the study is not clearly articulated and appears inconsistent between the Introduction and Discussion sections. In light of these issues, it is unclear whether the authors have successfully addressed—let alone fulfilled—the vaguely defined objectives.

Following comments are intended to support the authors in improving the clarity and scientific rigor of their manuscript, which has the potential to make a valuable contribution to the field:

Affiliations
-I am not familiar with the institution called Scientific advisor or President CIO3. These are more appropriate in the acknowledgements section or list of the credits attributed to each author - at the end of the article. 
Please correct affiliations and it is advisable to use professional e-mail addresses instead of the private ones. 

Abstract
-abbreviation for ozonized saline solution (O3SS) is not the same as in the introduction and results where number 3 is in subscript. Please ensure consistency in the terminology used.
-μg/NmL - what does N stands for in this measurement unit? Is it nano - in that case it should be formatted as a small letter. Please use SI units.

Introduction
-page 2: part of the text from line 50 till line 54 which ends with citing references 5-7 is basically same as the previous part of the text from line 45 till line 50 which ends with reference 4. Please revise those few sentences to avoid repetitive phrasing.

-page 2, line 73 - There is an unnecessary full stop ("We took advantage of.")

-In the last paragraph: the aim of the study is not clearly written. Also, in addition to the stated aim of the study, it would be advisable to list specific objectives that outline the purpose of the methods used. I am not entirely sure whether the authors intended to assess the anti-inflammatory and antioxidant effects of the solution itself or of the treated cells. Could this be clarified?
Also, it is stated that authors 'took advantage' of the cells. I'm not sure that's the most appropriate or professional expression. 

Results
-figure 1 - would not it be better to represent your data as percentage of cell viability. At the moment the reader cannot easily interpret the data - e. g. the ozone in concentration of 5 units at the same time increases number of live HUVEC cells and increases number of dead HUVEC cells??

-it would be useful to mention the reason for detection of hypochlorite

-page 4, line 96 - "h" is missing in the word "hypoclorite"

- page 4, line 99 - from the sentence "The antioxidant activity induced into the cells by the treatment with ozone..." it is not clear whether cells gain antioxidant activity or ozone has antioxidant effect on the cells. Please rephrase the text to make it clearer.

-figure 2 - please explain abbreviations used in the figure. O3 is not formatted correctly. In the methods it is mentioned that this method was qPCR, but in the description of the figure it is stated that qRT-PCR was used. These two methods differ. Please ensure consistency in the terminology used.

-figure 3 does not have clear and appropriate caption. It is just description of the methods and results. It would be good to state most important results from this analysis as a title of the figure. O3 is not formatted correctly. 

-figures 4 and 5 - same comment for qRT-PCR and O3 as for figure 2. 

-page 7 - name of the same cytokine is written in several forms: "IL-1b", "IL-1beta", ...

-figure 6, panel B - it is very hard to read the letters. Please increase quality of this part of the figure.

Discussion
-first paragraph - please rewrite the sentence about aim of the study to make it more clear

-page 9, line 198 - instead of "testes" should it be written "tested"?

-page 9, line 206 - periclinal studies?

-abbreviations used in the discussion should have been introduced at the first mention in the manuscript and not here

-I found the discussion section somewhat confusing—it's difficult to identify the authors' own results, as the text is dominated by comparisons with findings from other research groups.

-the manuscript would benefit from a clear statement of the study’s limitations.

Materials and methods
-page 12, line 326 - there is a typing error: it is stated that cells were seeded at a density of 5 × 105/well. Please correct the concentration.

-page 13 is blank

-page 14, part 4.4 Immunofluorescence - only in this part of the text you use ' instead of word for measurement unit of time (minute), and RT for room temperature. Please ensure consistency in the terminology used.

-page 14, line 353 - by 10% DMEM FBS you mean 10% Fetal Bovine Serum (FBS) in DMEM? 

-page 14, line 354 - abbreviation O3SS is not correctly formatted

-page 14, line 356 - how concentrated was PBS: 1×, 2×, 5×, 10×?

-page 14, line 356-357 - I am not sure I could follow this step (After 3 washes, TritonX-100 0.1% × 5′ was added). 
Please write methods clearly.

-page 14, lines 358-359 - this whole sentence in not clear: After 2 further washes in PBS, the cells were incubated with Primary Antibody 1:100 in PBS 01% BSA O.N. 4 °C for anti-ARG-1 AB-84248 (Immunological Sciences, Rome, It-359 aly) [15]15. What does O.N. stands for? What is ARG-1 AB-84248?

-page14, line 361 - antibody is not fluoresceinated - it is conjugated with fluorophore. Please write correctly name of the antibody and its producer: Rb CF488-a goat anti-Rb Ig(H+L).

-page14, lines 364-365 and lines 379-380 - please explain how exactly was fluorescence intensity quantified with ImageJ software. 

Author Response

The research topic is undoubtedly interesting and potentially valuable. However, the manuscript appears to have been prepared somewhat hastily, which may give the impression that the authors are relying on the peer review process to refine and improve the text. It is essential that manuscripts are clearly written and well-structured prior to submission, allowing reviewers to focus on evaluating the scientific merit rather than addressing issues related to presentation and clarity.

Furthermore, the aim of the study is not clearly articulated and appears inconsistent between the Introduction and Discussion sections. In light of these issues, it is unclear whether the authors have successfully addressed—let alone fulfilled—the vaguely defined objectives.

Following comments are intended to support the authors in improving the clarity and scientific rigor of their manuscript, which has the potential to make a valuable contribution to the field:

Affiliations
-I am not familiar with the institution called Scientific advisor or President CIO3. These are more appropriate in the acknowledgements section or list of the credits attributed to each author - at the end of the article. 
Please correct affiliations and it is advisable to use professional e-mail addresses instead of the private ones. 

Response: We thank the reviewer for their valuable comment regarding the authors' affiliations and email addresses.

We have revised the affiliations section accordingly. As correctly pointed out, the previous descriptions ("Scientific Advisor, Freelance" and "President CIO3") were more indicative of a role than a standard institutional affiliation. Following the reviewer's suggestion and in line with common practice for independent researchers, we have modified the affiliations to clearly state their status:

For Dr. [Gregorio Martínez-Sánchez], we have used the standard affiliation "Independent Researcher" along with the city and country.

For Dr. [Maggiorotti Maurizio], "CIO3" (Colegio Italiano di Ossigeno-Ozono Terapia) is a registered scientific society. To clarify its nature, we have rephrased the affiliation to "[...], Rome, Italy".

Regarding the email addresses, we appreciate the reviewer's concern. According to the MDPI's author guidelines, an institutional email is primarily required for the corresponding author to ensure communication and record stability. For co-authors, while institutional emails are preferred, personal emails are acceptable, especially for researchers who are retired, independent, or affiliated with societies that do not provide institutional email accounts. This is the case for the authors in question. However, to enhance professionalism, we have ensured that the personal emails provided are their dedicated and professional contact points for scientific correspondence.

The modifications made are detailed below. We believe the revised affiliations are now accurate and conform to the journal's standards.

Original

2 Martínez-Sánchez Gregorio, Scientific Advisor, Freelance, 60126 Ancona, Italy; gregorcuba@yahoo.it
...
4 Maggiorotti Maurizio, President CIO3, Rome, Italy; maurizio.maggiorotti@email.it

Modified:

2 Martínez-Sánchez Gregorio, Independent Researcher, 60126 Ancona, Italy; gregorcuba@yahoo.it
...
4 Maurizio Maggiorotti, Italian College of Oxygen-Ozone Therapy (CIO3), 00186 Rome, Italy; maurizio.maggiorotti@email.it

Abstract
-abbreviation for ozonized saline solution (O3SS) is not the same as in the introduction and results where number 3 is in subscript. Please ensure consistency in the terminology used.

We thank the referee for the suggestion. We have corrected it.

-μg/NmL - what does N stands for in this measurement unit? Is it nano - in that case it should be formatted as a small letter. Please use SI units.

The unit of measurement Nml, in reference to ozone (or other gases), means: "Normal milliliters" That is: milliliters measured under normal temperature and pressure conditions.

This unit is used to standardize the volume of gases, since gases change their volume depending on the temperature and pressure they are under.

Introduction
-page 2: part of the text from line 50 till line 54 which ends with citing references 5-7 is basically same as the previous part of the text from line 45 till line 50 which ends with reference 4. Please revise those few sentences to avoid repetitive phrasing.

We have corrected it according to your recommendation:

In this context, ozone has attracted significant interest due to its anti-inflammatory and antioxidant properties, placing it within the scope of complementary medicine—able to support the treatment of certain pathologies without fitting the definition of a pharmaceutical [4]. More recent research has emphasized its hormetic effects: while toxic at high concentrations, ozone can help counteract oxidative stress and inflammation when administered at low doses. These characteristics form the basis of ozone therapy [5–7].

-page 2, line 73 - There is an unnecessary full stop ("We took advantage of.")

Thank you. we removed the full stop.

-In the last paragraph: the aim of the study is not clearly written. Also, in addition to the stated aim of the study, it would be advisable to list specific objectives that outline the purpose of the methods used. I am not entirely sure whether the authors intended to assess the anti-inflammatory and antioxidant effects of the solution itself or of the treated cells. Could this be clarified?
Also, it is stated that authors 'took advantage' of the cells. I'm not sure that's the most appropriate or professional expression. 

Thank you! We have updated the text, clarifying the issues pointed out by the referee and the aim of our study.

The main aim of this preclinical study was to investigate the anti-inflammatory and antioxidant activity elicited by ozone  saline treatment within cells in the culture, and to evaluate the absence of cytotoxicity at different ozone concentrations. Specifically, we assessed the cellular response to a calibrated ozone/oxygen mixture delivered via ozonated saline solutions saturated with 1, 5, or 10 μg/Nml of O₃, focusing on both potential toxic effects and beneficial adaptive responses related to cellular antioxidant activation and inflammatory modulation. A very important aspect of the study was the determination of the minimum effective ozone dosage. To address these objectives, we employed two complementary in vitro models: primary endothelial cell cultures, used to evaluate possible cytotoxic effects that in vivo could manifest as phlebitis; and a murine microglial cell line, selected due to the growing interest in ozone therapy for neurodegenerative disorders such as Alzheimer’s and Parkinson’s diseases. In this context, our goal was to explore whether ozone exposure may modulate microglial activation states and stimulate endogenous antioxidant defenses. Through these models, we aimed to determine whether the ozone/oxygen mixture acts as a modulator of cellular antioxidant pathways and inflammatory responses, thereby supporting its potential use in diseases characterized by oxidative stress and inflammation.

, it is stated that authors 'took advantage' of the cells. I'm not sure that's the most appropriate or professional expression.

The expression “take advantage” is currently used in papers dealing with cell cultures to indicate the cellular models chosen for the experiments

Results
-figure 1 - would not it be better to represent your data as percentage of cell viability. At the moment the reader cannot easily interpret the data - e. g. the ozone in concentration of 5 units at the same time increases number of live HUVEC cells and increases number of dead HUVEC cells??

Thank you for the suggestion. We have modified the graph to show the values as percentages. The referee is right: we observed a simultaneous increase in live cells and an increase in dead cells. Based on these results, in subsequent experiments we chose the doses of 1 and 5 μg/ml, which proved to be more effective

-it would be useful to mention the reason for detection of hypochlorite

Response: We agree that explicitly stating the rationale for the hypochlorite detection assay will significantly improve the clarity of our methodological approach. We have now modified the relevant section in the Materials and Methods (subsection 4.6) to include a sentence that clearly explains the purpose of this specific measurement. The reason was to rule out hypochlorite formation as a potential confounding cytotoxic factor in our experiments.

Text on  (Page 14, lines 381-384) Was modified as

Detection of Reactive Chlorine Species (Hypochlorous Acid/Hypochlorite)

This measurement was performed as a crucial control to rule out that the biological effects observed in our experiments were mediated by the potential formation of hypochlorite (ClO⁻) as a byproduct of ozone dissolution, rather than by ozone itself. To quantify the levels of hypochlorous acid (HClO), we measured the concentration of the hypochlorite anion (ClO⁻) using a commercial Hypochlorite Detection Kit (Colorimetric) (ab219929, Abcam, Cambridge, UK). This method is based on a sensor that undergoes a specific reaction with ClO⁻ to generate a red-colored product, with absorbance measured at 555 nm. Given that HClO dissociates into H⁺ and ClO⁻ (pKa ≈ 7.5) and the kit detects the anionic form, the results provide an accurate indirect measurement of the total hypochlorous acid/hypochlorite content in the samples.

-page 4, line 96 - "h" is missing in the word "hypoclorite"

we have corrected it

- page 4, line 99 - from the sentence "The antioxidant activity induced into the cells by the treatment with ozone..." it is not clear whether cells gain antioxidant activity or ozone has antioxidant effect on the cells. Please rephrase the text to make it clearer.

Thank you! We have made the text clearer and more fluid.

The activation of cellular antioxidant pathways in response to treatment with ozone dissolved in 0.9% saline solution was evaluated by the analysis of several mRNAs, after 4 and 24 hours from treatment, as shown in Fig 2.The results confirmed the induction of an anti-oxidant response in the cells following the treatment with ozone dissolved in the saline solution

-figure 2 - please explain abbreviations used in the figure. O3 is not formatted correctly. In the methods it is mentioned that this method was qPCR, but in the description of the figure it is stated that qRT-PCR was used. These two methods differ. Please ensure consistency in the terminology used.

Thank you. We have modified the graph and added explanations of the abbreviations used in the captions: CTRL (control) LPS (lipopolysaccharide). We used qRT-PCR and corrected accordingly

-figure 3 does not have clear and appropriate caption. It is just description of the methods and results. It would be good to state most important results from this analysis as a title of the figure. O3 is not formatted correctly. 

As suggested by the referee, we have added a title to the caption (Ozone treatment reduces ROS expression in the LPS-stimulated BV2 cells) and formatted the figure with O3

-figures 4 and 5 - same comment for qRT-PCR and O3 as for figure 2. 

Done.

-page 7 - name of the same cytokine is written in several forms: "IL-1b", "IL-1beta", ...

Thank you. We have corrected it.

-figure 6, panel B - it is very hard to read the letters. Please increase quality of this part of the figure.

We apologize. We have improved the image quality.

Discussion
-first paragraph - please rewrite the sentence about aim of the study to make it more clear

Here is a revised, clearer, more fluid version:

Our study investigated the response of two distinct cellular models—BV2 microglial cells and HUVEC endothelial cells—following the exposure to ozone-saturated 0.9% saline solution (O₃SS), prepared by bubbling ozone for 20 minutes to achieve a final concentrations of 1, 5, or 10 µg/NmL.
The main objectives were: 1) to assess the potential cytotoxicity of O₃SS; and 2) to evaluate its antioxidant and anti-inflammatory activity at different ozone concentrations.

-page 9, line 198 - instead of "testes" should it be written "tested"?

We apologize for the confusion. We have corrected it.

-page 9, line 206 - periclinal studies?

We apologize for the confusion. We have corrected it.

-abbreviations used in the discussion should have been introduced at the first mention in the manuscript and not here

Done.

-I found the discussion section somewhat confusing—it's difficult to identify the authors' own results, as the text is dominated by comparisons with findings from other research groups.

Our study investigated the response of two distinct cellular models—BV2 microglial cells and HUVEC endothelial cells—following exposure to ozone-saturated 0.9% saline solution (O₃SS), prepared by bubbling ozone for 20 minutes to achieve final concentrations of 1, 5, or 10 µg/NmL. Our results demonstrate that the O3SS does not present toxic activity in itself in the tested concentration ranges. Previous studies have analyzed the possible reactions taking place after the addition of O3 to saline solutions ten ding with cytotoxicity. In this connection Razumovski et al. (2010) [17] established that ozone decomposition processes in NaCl aqueous solutions is not accompanied by formation of noticeable amounts of hypochlorites and chlorates. In addition, Peritiagyn demonstrated that the concentration of sodium hypochlorite in the O3SS was less than 0.001 g/mL [18]. We have confirmed this result by the hypochlorite detection kit. Moreover, it was demonstrated that ozonation of the saline solution eliminates traces of Bromine that exist in the normal pharmacological formulation of the saline [19]. The kinetics of ozone saturation in the saline solution were previously studied using spectrophotometric methods. At the concentrations used and with the same medical device and conditions, the ozone concentration in the solution stabilizes after 10 minutes of bubbling and corresponds to 10% of the initial concentration. Under these conditions, the formation of hypochlorous acid or hydrogen peroxide was not detected (Martínez Sánchez, Gregorio. (2020). Practical aspects in ozone therapy: Study of the ozone concentration in the ozonized saline solution, Ozone Therapy Global Journal, Vol. 10, nº 1, pp 55-68). Traces of Fe²⁺ in sodium chloride can, in principle, catalyze hydroxyl radical (•OH) generation through the Fenton reaction. However, this requires the presence of H₂O₂ or another suitable peroxide. In pure pharmaceutical-grade NaCl solution (0.9% saline), the European Pharmacopoeia sets a maximum Fe content of 2 ppm, and under standard ozonation conditions (1–10 µg/mL O₃), no detectable H₂O₂ is formed. Analytical studies confirm H₂O₂ levels remain far below the threshold necessary for Fenton chemistry. Therefore, while Fe²⁺ traces could act as catalysts in theory, in practice under these conditions the formation of hydroxyl radicals via the Fenton reaction is ruled out (Towards Reducing DBP Formation Potential of Drinking Water by Favouring Direct Ozone Over Hydroxyl Radical Reactions During Ozonation. De Vera GA, Stalter D, Gernjak W, et al. Water Research. 2015;87:49-58. doi:10.1016/j.watres.2015.09.007.) The proportion of O₃SS added to the DMEM medium was 1:20. Since DMEM contains a buffering system, significant pH variations after the addition of O₃SS were not expected.

Recently, systemic ozone application using O3SS was found to be well tolerated, with no serious adverse events reported in the preliminary study. The intervention, performed under controlled conditions and at low doses (3 μg/NmL), did not result in clinically significant toxicity or hematological abnormalities. Importantly, the study observed a transient increase in the survivor of circulating CD34+ cells following systemic ozone administration, suggesting a potential stimulatory effect on endothelial-hematopoietic stem/progenitor cell mobilization [20], perhaps explaining the proliferative effect observed on HUVEC cells in our results.

These findings are consistent with the established concept that low-dose ozone can induce a mild, controlled oxidative eustress, which in turn activates endogenous antioxidant and regenerative pathways without causing cellular damage [21,22]. The observed mobilization of CD34+ cells aligns with the literature describing ozone’s capacity to enhance tissue regeneration and modulate immune responses through redox bioregulation [7].

The redox regulatory effect of low doses ozone has been repeatedly described and has led to the inclusion of ozone treatments among the strategies to combat mitochondriopathies linked to high oxidative stress and often due to mitochondrial aging and dysfunction [8]. Several studies demonstrated that the O3SS-induced response is dependent on the activation of the transduction mechanisms of Nrf2 from cytoplasm to nucleus inducing antioxidant enzymes synthesis, such as SOD, CAT (catalase), and HO1 (heme oxygenase 1) among others [23–25]. In addition, recent preclinal studies demonstrate O3SS mitigates parthanatos after ischemic stroke. In both in vitro (SH-SY5Y cells exposed to H₂O₂) and in vivo (murine ischemic stroke) models, O3SS administration decreased oxidative stress and neuronal death [26]. Our results confirm the anti-oxidant activity of O3SS.

Intravenous infusion of a 5% glucose solution has made it possible to treat patients who could not be treated with the classical method, achieving similar results [27]. However, O3SS allows for precise control of ozone dosage and ensures immediate reaction of ozone with blood components, leading to the formation of redox-active messengers that trigger antioxidant, anti-inflammatory, and immunomodulatory responses.

Additionally, O3SS does not introduce an exogenous glucose load, thereby reducing the risk of metabolic complications such as hyperglycemia or electrolyte disturbances, which are documented risks with intravenous glucose infusions, especially in patients with metabolic comorbidities [28,29]. O3SS also minimizes the risk of infusion-related complications such as thrombophlebitis, which can occur with repeated glucose infusions [29].

Compared to major autohemotherapy (MAH), which involves ex vivo ozonation of a patient’s blood followed by reinfusion, administration of O3SS offers several practical advantages. O3SS is technically simpler, avoids direct blood handling, requires a lower calibre needle, and is more acceptable for patients who refuse blood manipulation due to religious or procedural concerns. Additionally, O3SS generally carries fewer legal implications since it does not involve extracorporeal blood processing.

However, O3SS has significant disadvantages as a lack of standardization. In general, the medical literature highlights that, while O3SS may be more convenient in certain settings, MAH remains the preferred method for systemic ozone therapy due to its superior efficacy, safety, and reproducibility when performed with appropriate protocols and individualized ozone concentrations. Notwithstanding, clinical evidence supports the efficacy and safety of both MAH and O₃SS infusions in diverse medical contexts. In the study by Makarov et al. (2017) [30], elderly patients undergoing rehabilitation for chronic cardiovascular and musculoskeletal conditions received a combination of MAH or O3SS and gravitational therapy. Over a follow-up period of up to 7 years, the combination of O3SS and gravitational therapy, resulted in the most significant reduction in the risk of disease progression and need for surgical intervention, as determined by Cox regression analysis.

In contrast, a retrospective analysis conducted by Andryushchenko et al. (2019) [31] (Andryushchenko, V.V.; Kurdil, N.V.; Struk, V.F.; Kalish, M.M. Actual Issues of the Practical Use of Parenteral Ozone Therapy in Emergency Medicine. Emergency Medicine 2019, 8, 121–127. https://doi.org/10.22141/2224-0586.8.103.2019.192383) 144 patients aged 17 to 72 years (58% women) hospitalized in an intensive care unit in Kyiv received parenteral ozone therapy as an adjunct to standard care for various acute and chronic conditions, including sepsis, trauma, burns, acute infections, diabetes, and peripheral vascular disease. Two main ozone therapy modalities were applied: intravenous O3SS (200 mL containing 0.48 mg of ozone) administered across 169 sessions, or MAH, involving 200–400 mL of autologous blood enriched with 1.8 mg of ozone, across 185 sessions. Clinically, both interventions were associated with improved blood oxygenation, enhanced rheological properties, activation of humoral immunity, and pain relief. Notably, no adverse events or complications were recorded in any of the 304 procedures, underscoring the safety of the approach. The study supports the practical utility of ozone therapy in emergency medicine and recommends further standardization and controlled clinical trials to validate and refine its use.

Microglial cells are considered among the main cellular players in the development of neuroinflammation underlying several chronic neurodegenerative diseases, including AD. For this reason, we choose the BV2 in vitro cellular model, analysing its behaviour after exposure to different amounts of ozone, in the presence or in the absence of LPS. Our results suggest that even with the lowest ozone concentrations used, microglia are polarized towards an anti-inflammatory phenotype. Ozone is able to down-regulate the expression of the pro-inflammatory cytokine IL-1β and the effect is greater after 24 h from treatment, while ozone alone is able to induce a greater expression of the anti-inflammatory cytokine IL-10, increasing its concentration even after exposure to LPS. Furthermore, the cellular markers of pro-inflammatory polarization (iNOS) are decreased, while the anti-inflammatory ones (Arg-1) are increased, as detected by real-time PCR and immunofluorescence.

Moreover, 4HNE obtained from PUFA after ozone addition, increases the expression and transactivation activity of PPARγ [34] and PPARγ were shown to reduces oxidative stress in brain tissue, improving mitochondrial function [35], reducing glial inflammation and Aβ levels in AD transgenic mouse models [36]. Our data confirm the ability of ozone to polarize microglial cells towards an anti-inflammatory phenotype, added to cells at low concentrations and dissolved in 0.9% NaCl saline solution.

According to the results, 0.9% saline solution exerts a hormetic effect on BV2 microglial cells, characterized by the activation of antioxidant and anti-inflammatory pathways without inducing cytotoxicity at clinically relevant concentrations. The observed upregulation of Nrf2 and SOD1 mRNA at 4 and 24 hours indicates that ozone triggers the Keap1/Nrf2-dependent antioxidant response, leading to increased transcription of genes encoding key antioxidant enzymes such as SOD, which is consistent with the established mechanism of low-dose ozone as a eustress inducer [37]. This activation enhances the cellular capacity to ROS, as evidenced by the reduction in ROS levels following LPS stimulation.

The modulation of inflammatory markers further supports the immunoregulatory role of ozone. Downregulation of iNOS mRNA and upregulation of Arg-1 mRNA reflect a shift from a pro-inflammatory (M1-like) to an anti-inflammatory (M2-like) microglial phenotype, aligning with the literature showing that low-dose ozone suppresses pro-inflammatory mediators and promotes anti-inflammatory gene expression [38,39]. This dual action, attenuation of oxidative stress and reprogramming of microglial activation, underpins the therapeutic rationale for ozone therapy in neuroinflammatory and neurodegenerative contexts.

-the manuscript would benefit from a clear statement of the study’s limitations.

Thank you for the suggestion. We have added the limitations section after the conclusions.

  1. Limitations

Although our results provide promising evidence of the antioxidant and anti-inflammatory potential of O₃SS, certain limitations must be acknowledged. First, the study was conducted exclusively on cell cultures in vitro (murine microglia BV2 and human endothelial cells HUVEC) which, although informative, do not fully replicate the complexity of physiological environments in vivo. Furthermore, the effects of repeated or chronic exposure to O₃SS were not evaluated, and the short-term cellular responses observed here may not reflect long-term outcomes. Finally, although we observed modulation of antioxidant and inflammatory markers, the molecular mechanisms underlying these effects, such as specific signaling pathways or interactions with redox-sensitive transcription factors, require further investigation. Clinically, ozone therapy (via O₃SS or MAH) has shown efficacy, without serious adverse effects reported. While both methods have demonstrated safety and potential efficacy, current limitations include variability in ozone dosing, delivery protocols, and lack of universal guidelines. Future in vivo studies are essential to validate these findings, determine pharmacokinetics, and establish safe and effective dosing protocols for potential clinical applications.

Materials and methods
-page 12, line 326 - there is a typing error: it is stated that cells were seeded at a density of 5 × 105/well. Please correct the concentration.

Thank you, we have corrected the typo.

-page 13 is blank

Done.

-page 14, part 4.4 Immunofluorescence - only in this part of the text you use ' instead of word for measurement unit of time (minute), and RT for room temperature. Please ensure consistency in the terminology used.

Done

-page 14, line 353 - by 10% DMEM FBS you mean 10% Fetal Bovine Serum (FBS) in DMEM? 

Yes! we have clarified the sentence in the text. Thank you!

-page 14, line 354 - abbreviation O3SS is not correctly formatted

Done.

-page 14, line 356 - how concentrated was PBS: 1×, 2×, 5×, 10×?

We have added the concentration of 1x PBS to the text.

-page 14, line 356-357 - I am not sure I could follow this step (After 3 washes, TritonX-100 0.1% × 5′ was added). 
Please write methods clearly.

we clarified in the text:

After three washes, cells were permeabilized with 0.1% Triton X-100 for 5 minutes.

-page 14, lines 358-359 - this whole sentence in not clear: After 2 further washes in PBS, the cells were incubated with Primary Antibody 1:100 in PBS 01% BSA O.N. 4 °C for anti-ARG-1 AB-84248 (Immunological Sciences, Rome, It-359 aly) [15]15. What does O.N. stands for? What is ARG-1 AB-84248?

We hope this makes it clearer:

After two additional washes in PBS, the cells were incubated overnight at 4 °C with the rabbit primary antibody anti-Arginase-1 (AB-84248, Immunological Sciences, Rome, Italy) diluted 1:100 in PBS containing 0.1% BSA.

-page14, line 361 - antibody is not fluoresceinated - it is conjugated with fluorophore. Please write correctly name of the antibody and its producer: Rb CF488-a goat anti-Rb Ig(H+L).

Thank you, we have rewritten the sentence: After 3 washes in PBS, the cells were incubated with secondary antibody conjugated with fluorophore CF488A-conjugated goat anti-rabbit IgG (H+L) (Biotium, Fremont, CA, USA)

-page14, lines 364-365 and lines 379-380 - please explain how exactly was fluorescence intensity quantified with ImageJ software. 

Using ImageJ software, version 1.48, the fluorescence intensity in the FITC channel was measured for each field and expressed as the mean fluorescence per cell by dividing the total signal by the number of cells measured by DAPI nuclei staining

Reviewer 4 Report

Comments and Suggestions for Authors
  • The high and low concentration ranges must be specified in the following paragraph: “The gas has hormetic properties: harmful at high concentrations, it is beneficial at low concentrations as it stimulates cellular antioxidant defenses, capable of counteracting oxidative stress, as detailed below”.
  • Idem for the following paragraph: “More recent studies have high-lighted the hormetic properties of ozone: toxic at high concentrations, it is instead supportive in counteracting oxidative stress as well as inflammation at low doses”.
  • The term in vitro should be in italics.
  • In the introduction, state whether ozone therapy has previously been used to treat phlebitis or neurodegenerative diseases such as Alzheimer's. If not, explain why you would expect this therapeutic strategy to be effective.
  • The first paragraph of the results section must be accompanied by its bibliographic reference: “Our preclinical study aims at evaluating the efficacy of low doses of O3SS added to cultures of BV2 microglial cells in carrying out an anti-oxidant and anti-inflammatory activity, after having evaluated their possible cytotoxic activity”.
  • Figure 1 shows the meaning of the acronym CTRL and the number 03, which precedes the ozone concentration values on the abscissa axis of the graphs.
  • Can you explain the difference in the p values marked with * and # in Figure 1?
  • What is the basis for detecting hypochlorite in the tests? The results of these tests are shown in Figure 1.
  • Fundamentally because it determined the expression of Nrf2 mRNA and SOD1 mRNA in BV2 cells.
  • Please could you explain what the number 03 means that appears before the concentration values in Figure 3?
  • Give the definitions of the acronyms iNOS and Arg-1.
  • However, the objectives of this work also proposed carrying out studies on epithelial cells to understand the possible application of ozone therapy in phlebitis, yet the experiments were only carried out on BV2 cells. So why were these tests not also carried out on epithelial cells?
  • The authors make these statements as if they had carried out these determinations in this work. 'However, this study also aimed to investigate the potential application of ozone therapy in phlebitis by conducting experiments on epithelial cells, yet these experiments were only carried out on BV2 cells´. Why were these tests not also carried out on epithelial cells?' However, this is not the case. Please clarify this.
  • The cytotoxicity assays are expressed as a number of cells, rather than as a percentage of viability. Could you please express the results in Figure 1 as a percentage of cell viability? Otherwise, the statement: 'O3SS is able to exert a proliferative effect on microglia and endothelial cell cultures. We can therefore conclude that O3SS does not exhibit toxic activity at the tested concentrations' is unfounded.
  • What methodology did the authors adopt in order to ascertain the true ozone concentration in the saline solution?

Author Response

  • The high and low concentration ranges must be specified in the following paragraph: “The gas has hormetic properties: harmful at high concentrations, it is beneficial at low concentrations as it stimulates cellular antioxidant defenses, capable of counteracting oxidative stress, as detailed below”.

Response: We thank the reviewer for this excellent suggestion. We agree that specifying the concentration ranges will significantly improve the clarity and scientific rigor of our introduction. We have now modified the sentence on Page 2, Line 18 to include the well-established therapeutic concentration range for medical ozone (between 1 and 80 µg/mL). The revised text now clearly states that beneficial, hormetic effects are observed within this range, while harmful effects are associated with concentrations exceeding it.

Original text in page 2 line 45

“The gas has hormetic properties: harmful at high concentrations, it is beneficial at low concentrations as it stimulates cellular antioxidant defenses, capable of counteracting oxidative stress, as detailed below [3].”

Was modified as:

“The biological effects of ozone are concentration-dependent and follow a hormetic pattern. While high concentrations (>80 µg/mL) are toxic and harmful, low concentrations within the well-defined therapeutic window (1-80 µg/mL) are beneficial. This beneficial effect is mediated through the stimulation of cellular antioxidant defenses, which are ultimately capable of counteracting oxidative stress, as detailed below [3].”

  • Idem for the following paragraph: “More recent studies have high-lighted the hormetic properties of ozone: toxic at high concentrations, it is instead supportive in counteracting oxidative stress as well as inflammation at low doses”.

Response: We thank the reviewer for their consistency in pointing out the need for quantitative precision. As we did in the previous paragraph, we have now modified this sentence to specify the concentration ranges that define the toxic and beneficial effects of ozone. The revised text now clearly states that the supportive, hormetic effects are observed within the therapeutic concentration range of 1-80 µg/mL, while toxic effects are associated with concentrations above this range.

Original text in page 2 line 50

“More recent studies have high-lighted the hormetic properties of ozone: toxic at high concentrations, it is instead supportive in counteracting oxidative stress as well as inflammation at low doses”.

Was modified as:

More recent studies have highlighted the dual, dose-dependent nature of ozone: harmful at high concentrations (>80 µg/mL) but beneficial at low doses within the established therapeutic window (1-80 µg/mL), where it supports the counteraction of oxidative stress and inflammation

  • The term in vitro should be in italics.

Thank you for your observation. We have now italicized all occurrences of in vitro throughout the manuscript, as requested.

  • In the introduction, state whether ozone therapy has previously been used to treat phlebitis or neurodegenerative diseases such as Alzheimer's. If not, explain why you would expect this therapeutic strategy to be effective.

As far as phlebitis is concerned we reply later. The ozone autohemotherapy has been successfully employed for the treatment of acute cerebral infarction (Cheng H, Lu R, Du J, Zhao X, Zhuang Z. A clinical study on ozone autohemotherapy for the treatment of acute ischemic stroke. Front Med (Lausanne). 2025 Jul 25;12:1595568. doi: 10.3389/fmed.2025.1595568. PMID: 40786087; PMCID: PMC12331603.) Ozone preconditioning demonstrated a neuroprotective effect by increasing NRF2 nuclear translocation and the expression of SLC7A11 and GPX4 in a cerebral ischemia/reperfusion injury rat model (Zhu F, Ding S, Liu Y, Wang X, Wu Z. Ozone-mediated cerebral protection: Unraveling the mechanism through ferroptosis and the NRF2/SLC7A11/GPX4 signaling pathway. J Chem Neuroanat. 2024 Mar;136:102387. doi: 10.1016/j.jchemneu.2023.102387. Epub 2024 Jan 3. PMID: 38182039.). In addition ozone therapeutic value was shown in treating central nervous system diseases (especially ischemic stroke and Alzheimer's disease); as a matter of fact herapeutic value in treating central nervous system diseases (especially ischemic stroke and Alzheimer's disease) and the toxic effects of ozone, we find that ozone inhalation and a lack of antioxidants or excessive exposure lead to harmful impacts. However, with adequate antioxidants, ozone can transmit oxidative stress signals, reduce inflammation, reduce amyloid β peptide levels, and improve tissue oxygenation. .  ozone can transmit oxidative stress signals, reduce inflammation, reduce amyloid β peptide levels, and improve tissue oxygenation. (Zhang X, Wang SJ, Wan SC, Li X, Chen G. Ozone: complicated effects in central nervous system diseases. Med Gas Res. 2025 Mar 1;15(1):44-57. doi: 10.4103/mgr.MEDGASRES-D-24-00005. Epub 2024 Oct 2. PMID: 39436168; PMCID: PMC11515058.).A further study concluded that  Oxygen-ozone therapy would contribute to improving the MS patients by elevating the Treg cell responses. (Tahmasebi S, Qasim MT, Krivenkova MV, Zekiy AO, Thangavelu L, Aravindhan S, Izadi M, Jadidi-Niaragh F, Ghaebi M, Aslani S, Aghebat-Maleki L, Ahmadi M, Roshangar L. The effects of oxygen-ozone therapy on regulatory T-cell responses in multiple sclerosis patients. Cell Biol Int. 2021 Jul;45(7):1498-1509. doi: 10.1002/cbin.11589. Epub 2021 Mar 26. PMID: 33724614. In addition it is known that brain microenvironment is shaped by glial cells receiving information by neurons, vascular elements as well as by peripheral immune signals , providing the rationale for the use of ozone for the treatment of central nervous system disorders (Müller L, Di Benedetto S, Müller V. The dual nature of neuroinflammation in networked brain. Front Immunol. 2025 Aug 20;16:1659947. doi: 10.3389/fimmu.2025.1659947. PMID: 40909282; PMCID: PMC12404926.)

  • The first paragraph of the results section must be accompanied by its bibliographic reference: “Our preclinical study aims at evaluating the efficacy of low doses of O3SS added to cultures of BV2 microglial cells in carrying out an anti-oxidant and anti-inflammatory activity, after having evaluated their possible cytotoxic activity”.

Response: We thank the reviewer for their careful reading of our manuscript. We would like to kindly clarify that the sentence in question does not report a finding from the literature but rather states the primary objective of our present study, as it was conceived and conducted by us. Therefore, it is an original aim and not a claim that requires a bibliographic citation. This introductory sentence serves to remind the reader of the goal of the experiments whose results are presented immediately afterward in the same section.

For the sake of clarity and to avoid any potential confusion for the reader, we have slightly rephrased the sentence to make its self-referential nature even more explicit. The modified version now reads: "In the present preclinical study, we aimed to evaluate..."

We hope this modification addresses the reviewer's concern appropriately.

Original text in Page 3 line 80:

“Our preclinical study aims at evaluating the efficacy of low doses of O3SS added to cultures of BV2 microglial cells in carrying out an anti-oxidant and anti-inflammatory activity, after having evaluated their possible cytotoxic activity”.

Was modified as:

In the present preclinical study, we aimed to evaluate the efficacy of low doses of O3SS added to cultures of BV2 microglial cells in carrying out an anti-oxidant and anti-inflammatory activity, after having evaluated their possible cytotoxic activity”.

  • There is no need of reference quotation since the paragraph refers to the results presented in this study (this paper).
  • We thank the referee for the recommendation. We have added the word “control” to the caption and have improved the notation of the ozone symbol O3 in the graph to make it clearer.

  • Can you explain the difference in the p values marked with * and # in Figure 1?

Thank you for your comment. In Figure 1, the symbols * and # indicate statistical comparisons against two different reference groups:

The asterisk (*) denotes a statistically significant difference compared to the live cell in the control group, indicating changes relative to viable cells.

The hash (#) indicates a statistically significant difference compared to the dead cell in the control group, highlighting differences relative to non-viable cells.

This distinction allows us to separately assess the effects of the treatment on live versus dead cell populations. We apologize if this was not clearly explained in the figure legend, and we revise it accordingly.

  • What is the basis for detecting hypochlorite in the tests? The results of these tests are shown in Figure 1.

Response: We thank the reviewer for this question, which allows us to clarify this important point. We apologize for any confusion caused. The results of the hypochlorite detection assay are not displayed in Figure 1. Figure 1 exclusively shows the results of the Trypan Blue cytotoxicity assay for BV2 and HUVEC cells.

The hypochlorite detection was performed as a separate, crucial control experiment to rule out that the observed cytotoxicity (shown in Figure 1) was caused by the potential formation of hypochlorite (ClO⁻) as an artifact of the ozone (O₃) bubbling process in the solution, rather than by O₃ itself. The basis for this detection was the use of the Hypochlorite Detection Kit (Colorimetric) (ab219929, Abcam), as detailed in the Materials and Methods section (subsection 4.6). This kit uses a selective sensor that reacts specifically with hypochlorite to produce a colorimetric signal.

The results of this specific control experiment are described in the text of the Results section. To prevent future misunderstanding, we have modified the relevant sentence to be more precise and to explicitly state the purpose of the assay.

Original text in page 4 line 94-98:

"Moreover using hypoclorite detection kits we measured a hypochlorite concentration of 0.001%, similar to the value found in the culture medium, with all O3 concentrations used. It is to note that such a value that corresponds to the detection limit of the kit."

Was modified as:

To rule out that the cytotoxic effects observed in Figure 1 were due to the formation of hypochlorite (ClO⁻) during the ozonation process, we measured its concentration in the ozonated solutions using a specific colorimetric kit. The measured hypochlorite concentration was negligible (0.001%) across all O₃ concentrations tested, a value similar to that found in the culture medium alone and corresponding to the detection limit of the assay kit. This result indicates that hypochlorite formation was not a significant factor contributing to the cytotoxicity of the O₃ solutions.

  • Fundamentally because it determined the expression of Nrf2 mRNA and SOD1 mRNA in BV2 cells.

Response: We thank the reviewer for the opportunity to elaborate on the fundamental rationale behind measuring Nrf2 and SOD1 mRNA expression in BV2 microglial cells.

The primary objective of this specific part of our study was to corroborate whether the well-established molecular mechanism of action of ozone, previously described in other biological systems, was also engaged in our experimental model using ozonated saline solution (O3SS) on microglial cells.

It is widely recognized that the therapeutic effects of low doses of ozone are predominantly mediated through the activation of the Nuclear factor erythroid 2-related factor 2 (Nrf2) pathway. Upon activation, Nrf2 translocates to the nucleus and binds to the Antioxidant Response Element (ARE), initiating the transcription of a battery of cytoprotective and antioxidant genes, including superoxide dismutase (SOD).

Therefore, we specifically selected:

  1. Nrf2 mRNA: To directly assess the activation of this master transcriptional regulator at the gene expression level in response to our O3SS treatments.
  2. SOD1 mRNA: To evaluate the downstream functional outcome of Nrf2 pathway activation, as SOD1 is a key antioxidant enzyme whose upregulation is a classic hallmark of a successful Nrf2-mediated antioxidant response.

In essence, these measurements were not arbitrary; they were chosen as fundamental molecular markers to provide mechanistic insight and confirm that the observed biological effects (reduction in oxidative stress and inflammation) in our BV2 cell model were indeed triggered through the canonical Nrf2 pathway, thus aligning our findings with the established hormetic theory of ozone action. This approach allows us to move beyond merely observing a phenomenological effect and towards understanding the underlying molecular mechanism in our experimental setup.

Original text in Page 4 line 101

We evaluated the expression of Nrf2

Was modified as:

To investigate the molecular mechanism underlying the antioxidant effects of O3SS, we assessed the activation of the Nrf2 pathway by measuring the expression of Nrf2 and SOD1 mRNA

  • Please could you explain what the number 03 means that appears before the concentration values in Figure 3?

Thank you for your observation. The number "O3" appearing before the concentration values in Figure 3 refers to ozone. To improve clarity, we have updated the figure labels and legends to explicitly indicate the chemical formula "O₃" so that it is clear that the values correspond to ozone concentrations.

  • Give the definitions of the acronyms iNOS and Arg-1.

Done.

  • However, the objectives of this work also proposed carrying out studies on epithelial cells to understand the possible application of ozone therapy in phlebitis, yet the experiments were only carried out on BV2 cells. So why were these tests not also carried out on epithelial cells? The authors make these statements as if they had carried out these determinations in this work. 'However, this study also aimed to investigate the potential application of ozone therapy in phlebitis by conducting experiments on epithelial cells, yet these experiments were only carried out on BV2 cells´. Why were these tests not also carried out on epithelial cells?' However, this is not the case. Please clarify this.

Cytotoxicity tests were conducted on endothelial cells because irritant damage induced by treatments with toxic substances, for example inadequate concentrations of ozone, can cause phlebitis. Therefore, cytotoxicity tests were conducted to exclude the possibility of causing phlebitis after treatment with ozone.

The experiments were correctly performed on endothelial cells that form the intima of the blood vessels wall. Epithelial cells are not present in the vessels nor are they involved in phlebitis

  • The cytotoxicity assays are expressed as a number of cells, rather than as a percentage of viability. Could you please express the results in Figure 1 as a percentage of cell viability? Otherwise, the statement: 'O3SS is able to exert a proliferative effect on microglia and endothelial cell cultures. We can therefore conclude that O3SS does not exhibit toxic activity at the tested concentrations' is unfounded.

Thank you for the suggestion. We have modified the graph to show the values as percentages.

  • What methodology did the authors adopt in order to ascertain the true ozone concentration in the saline solution?

Response: We thank the reviewer for this critical question, which allows us to clarify our methodology for ozone quantification.

The concentration values of 1, 5, and 10 µg/mL stated in the manuscript for the treatments correspond to the output concentration set on the WeZONO ozone generator (i.e., the concentration of the O₂/O₃ gas mixture produced).

The methodology to ascertain the true final concentration of ozone dissolved in the saline solution was established in a prior, dedicated calibration study using the identical equipment and protocol (same device, vial type, volume, needle size, bubbling time, and gas flow).

That previous study, validated by standardized iodometric titration, determined that the real dissolved ozone concentration in the saline solution consistently represents 10% of the output concentration set on the generator. Therefore, for example, a generator output set at 10 µg/mL results in a final dissolved ozone concentration of approximately 1 µg/mL in the saline solution.

We applied this pre-validated 10% conversion factor to all our experiments. The full methodological details and validation data are described in:
Martínez Sánchez, G. (2020). Practical aspects in ozone therapy: Study of the ozone concentration in the ozonized saline solution. Ozone Therapy Global Journal, 10(1), 55-68.

We will revise the manuscript to express this with absolute clarity and avoid any potential misunderstanding.

Original text on page 11 line 314-319

"Treatment was performed by mixing the culture medium with saline solution (1:20 dilution). Ozone gas (O₃) was added to the wells after being dissolved in 0.9% saline solution (Eurospital, Trieste, Italy) by bubbling for 20 minutes at final concentrations of 1, 5, or 10 µg/NmL. The ozone-saline mixture was prepared using the WeZONO device (Deva Med. Italy)."

Was modified as:

Treatments were performed by mixing the ozonized saline solution with culture medium at a 1:20 dilution. The ozonized saline was prepared by bubbling an O₂/O₃ gas mixture (output from the WeZONO device (Deva Med. Italy) set at 1, 5, and 10 µg/mL) into 0.9% saline solution (Eurospital, Trieste, Italy) for 20 minutes. Based on a previously established and validated methodology [Ref], the actual concentration of ozone dissolved in the saline solution is 10% of the generator output value, resulting in concentrations of 0.1, 0.5, and 1 µg/mL prior to dilution.

Ref. Martínez Sánchez, G. (2020). Practical aspects in ozone therapy: Study of the ozone concentration in the ozonized saline solution. Ozone Therapy Global Journal, 10(1), 55-68.

Round 2

Reviewer 3 Report

Comments and Suggestions for Authors

I consider this study to be a valuable contribution to the current body of knowledge, and I would like to see it published. Thank you to the authors for accepting the suggestions. Some of the proposed changes have been incorporated into the manuscript. However, in certain areas, the authors have not succeeded in improving the clarity of the text or maintaining consistency in the use of abbreviations.

-The abbreviation for ozone saline solution is still not used consistently. When referring to ozone, please use the subscript “3” (i.e., O₃), and when referring to the ozone saline solution, please consistently use the abbreviation you have chosen (with or without the subscript). Since it seems that the commonly used abbreviation is O3SS, kindly ensure this is applied uniformly throughout the manuscript.

-When a unit of measurement is commonly used in a specific form, please ensure that this form is used consistently throughout the manuscript wherever applicable.

-In the Abstract, the aim of the study is clearly stated: evaluation of the cytotoxicity, antioxidant response, and immunomodulatory effects of O₃SS on murine microglial (BV2) and human endothelial (HUVEC) cells. These aspects are also appropriately addressed in the Discussion section.
However, this objective is not clearly articulated in the Introduction. The newly added paragraph (lines 96–135) includes repeated statements, some content more suitable for the Methods section, and parts that would be better placed earlier in the Introduction, prior to stating the study’s aim.
Kindly revise this section to clearly and explicitly state the main aim and specific objectives of the study.

-To improve the clarity and flow of English in lines 52–53, I kindly suggest the following revision:
“Due to its strong oxidizing properties, it was widely used during the World Wars for wound treatment and disinfection, and it continues to be employed today for water sanitation.”

-In the Introduction, lines 57–58 are repeated later in the text (lines 71–74). To avoid redundancy, I kindly recommend removing the repeated content from lines 71–74.

-Lines 87–88 contain the following reference: (Schwartz Tapia, A. 87 ISCO3/MET/00/21 Ozonized Saline Solution (O₃SS). 2025. Available online: https://isco3.org/officialdocs/) [10]. And lines 462-464: International Scientific Committee of Ozone Therapy (ISCO3). (2019). Guidelines and Recommendations for Medical Professionals Planning to Acquire a Medical Ozone Generator. ISCO3. Retrieved from https://isco3.org/wp-content/up-464 loads/2015/09/Generadores-ISCO3-SOP-DEV-01-01-2019.). 
Kindly format these references according to the Instructions for Authors provided by the journal. Please ensure that all elements (author name, title, year, source, and access link) are presented in the correct style and order.

-In line 98, the phrase “mouse model of microglia” implies that mice were used in the study, which is not the case. Since the research was conducted using murine microglial cell lines, please revise this sentence to accurately reflect the experimental model. Kindly use the correct and complete name of the cell line (e.g., BV2 murine microglial cells).

-Please ensure that all figure descriptions follow a uniform format - e.g. regarding statistical comparisons, choose one consistent phrasing—either "statistical comparisons against live cells" or "vs. live cells". I would recommend the latter, as it keeps the figure captions concise.

-After the revision, Figure 3 and Figure 5 appear to be of lower quality. Could the resolution be improved?

-The text of the manuscript overlaps with Figure 3. Please correct this formatting issue.

-The caption for Figure 3 is still incomplete. After the title, please include the following (instead of the current statement): Quantification of the median fluorescence of the cell-permeable probe 2',7'-dichlorodihydrofluorescein diacetate using ImageJ. The data are expressed as...

-Suggested caption for Figure 6: Evaluation of mRNA expression of IL-1β (A) and IL-10 (B) assessed by qRT-PCR. ...

-Latin expressions such as in vitro, in vivo, ex vivo, etc., should be italicized throughout the manuscript.

-In the Discussion section, there are several unexplained abbreviations (e.g., 4HNE, PUFA, PPARγ, Aβ, AD). Please define these upon first use.

-Line 503 contains a time duration expressed using both the apostrophe symbol (') and the word minute. To ensure clarity and consistency, please choose only one format for indicating time throughout the manuscript. There is an apostrophe also in line 506.

-Kindly revise the antibody name to the correct format: goat anti-rabbit IgG (H+L) conjugated with CF488A. Please do not forget to state the manufacturer and catalog number.

-line 534 - please cite ImageJ where it is first metioned in the methods - line 517.

-Instead of starting the sentence with “This measurement”, I kindly suggest rephrasing it to:
“The measurement of the hypochlorite anion was performed...”

-In reference number 10, please add the following information:
“Available online at: https://isco3.org/officialdocs/”

Reviewer 4 Report

Comments and Suggestions for Authors

The authors' responses to the reviewer's suggestions and observations were satisfactory.